# Assessment of spectral UV radiation at Marambio Base, Antarctic Peninsula

Klára Čížková[1,2], Kamil Láska[1], Ladislav Metelka[2], Martin Staněk[2]

[1]Department of Geography, Faculty of Science, Masaryk University, Brno, 611 37, Czech Republic

[2]Solar and Ozone Observatory, Czech Hydrometeorological Institute, Hradec Králové, 500 08, Czech Republic

*Correspondence to*: Klára Čížková (cizkova.klara@mail.muni.cz)

**Abstract.** This study aims to assess the dependence of spectral UV radiation on different atmospheric and terrestrial factors, including solar zenith angle, ozone, and cloud cover, in the southern polar environment. For this purpose, 23260 spectra (300–363 nm), obtained by the B199 Mk-III Brewer spectrophotometer at Marambio Base, Antarctic Peninsula Region, over the

period 2010–2020, were studied. A neural network model was developed to investigate the effects of the explanatory variables at 127 wavelengths in the interval 300–363 nm, with a 0.5 nm sampling interval. Solar zenith angle (SZA) proved to be the most important parameter, followed by cloud cover, total ozone column (TOC), and surface albedo. The relative SZA effect is greatest at the shortest wavelengths, where a 1° decrease in SZA results in a 6–18 % increase in UV irradiance (305 nm). TOC affects particularly the short wavelengths below approximately 320–325 nm, when for example at 305 nm, a 10 DU

decrease in TOC causes a 7–13 % increase in UV irradiance. The large-scale ozone holes (e.g., in 2011–2012, 2014–2015, 2018–2019) caused the spectral UV irradiance at very short wavelengths peak in spring, whereas in other seasons (e.g., 2010–2011, 2012–2013), the maxima at all wavelengths were recorded in summer (November to January). Absorption of UV radiance by ozone affected also the temporal distribution of very high spectral UV irradiances (i.e. highest 10 % of the distribution), when at 305 nm, they were observed both in spring and summer months, and at 340 nm, they occurred mostly in

summer. The effect of cloud cover was strongest near the fully cloudy sky, and in summer months, when the Antarctic clouds tend to be thickest.

## 1 Introduction

Solar UV radiation has been a source of scientific interest since its discovery in 1801 (Hockberger, 2002). The biological effects of UV radiation, such as its erythemal effects in humans, were first described in the late 19th century, followed by

medical research in the 20th century (Diffey, 1980). Since then, many biological effects have been attributed to UV radiation in humans (e.g., Czerwińska and Krzyścin, 2019; Holick, 2016; Young et al., 2021a), other organisms or even entire ecosystems (e.g., Barnes et al., 2022; Young et al., 2021b).

As stratospheric ozone is the most important gas affecting the absorption of UV radiation in the atmosphere, preventing as much as 97–99 % of incident short-wavelength UV irradiance from reaching the surface (e.g., McKenzie et al., 2007), UV

radiation research received the greatest attention after the discovery of the Antarctic ozone hole in 1985 (Chubachi, 1985; Farman et al., 1985). Since then, many efforts have been made to reduce ozone depletion, most notably through the passing of the Montreal Protocol in 1987 and subsequent amendments to this landmark treaty (Velders et al., 2007). The recent positive stratospheric ozone trends in southern polar regions (e.g., McKenzie et al., 2019; Pazmiño et al., 2018; Solomon et al., 2016) contributed to a decrease (albeit statistically insignificant) in UV irradiance recorded at various Antarctic sites (Bernhard and Stierle, 2020). However, the ozone hole still keeps forming each Antarctic spring, increasing the UV irradiances and threatening Antarctic ecosystems (e.g., Cordero et al., 2022). UV radiation assessment is, therefore, still an important scientific task in both hemispheres, including for example the Arctic ozone loss in spring 2020 (e.g., Bognar et al., 2021).

The solar UV spectrum is according to the World Health Organization standard (WHO et al., 2002) divided into three bands: UVC (100–280 nm), UVB (280–315 nm), and UVA (315–400 nm). However, this division is semi-arbitrary, as there is no clear, physically defined transition between UVA and UVB bands (e.g., Diffey, 1990; Juzeniene et al., 2011). The shape of the solar UV spectrum is given mostly by the incident solar spectral irradiance at the top of the atmosphere and the wavelength-dependent absorption of spectral UV irradiance by various atmospheric gases (e.g., Kerr and Fioletov, 2008). The extraterrestrial solar spectrum has long been described, and the strong absorption at certain wavelengths has been explained by the Fraunhofer lines, for example the strong iron line at about 358 nm or the nitrogen hydrogen line at 336 nm (Stair, 1951). The major atmospheric absorbers in the UVB region are ozone, nitrogen dioxide and sulphur dioxide, but due to the difference in their respective abundances, ozone reduces the surface UV irradiance 200 to 300 times more than the other two (DeLuisi, 1997). The UVB region is mostly affected by the Huggins absorption band, which spreads between approximately 310 and 370 nm and exhibits a sequence of wave-like structures (Qu et al., 2004).

Spectral UV observations require a quality instrument with high sensitivity, satisfactory wavelength resolution and large stray light rejection (Webb, 1991). Such instruments only became available in the 1990s. Since then, solar UV spectra have been monitored at various locations for several decades (e.g., McKenzie et al., 1993; Webb, 1991), especially after the introduction of the double monochromator Brewer Mk-III spectrophotometer (Bais, 1997). However, the assessment of spectral solar UV irradiance in the Antarctic remains rather infrequent. In most studies from Antarctica, UV irradiance is integrated using various action spectra, most commonly the erythemal action spectrum, or several selected wavelengths or their intervals are evaluated. Nevertheless, continuous spectral measurements exist in Antarctica for example at the U.S. research stations Amundsen-Scott at South Pole (e.g., Bernhard et al., 2004; 2008; 2010), McMurdo (Bernhard et al., 2006; 2010), and Palmer Station (Bernhard et al., 2005; 2010). Apart from these works, the entire solar UV spectrum has been mentioned only in few instances, such as the case studies from Union Glacier Camp by Cordero et al. (2014) or Escudero Station by Cordero et al. (2013), who included short observation campaigns.

This study aims to contribute to broaden the knowledge of spectral UV irradiance in Antarctica by assessing the 10 year long time series of spectral UV irradiances recorded at Marambio Base in the Antarctic Peninsula region. We focused on UV climatology and year to year differences, extreme values, as well as on the factors affecting the spectral UV irradiance at individual wavelengths. We studied the effects of solar zenith angle (SZA), total ozone column (TOC), and cloud cover using

artificial neural network (ANN) modelling, which allowed us to focus on key factor and linkages between the southern polar

vortex and spectral UV irradiance reaching the site. The paper is organized in two main sections: Section 2 gives the data and methods used in the study. The study site is specified in Section 2.1, the instrumentation in Section 2.2, and other datasets (SZA, TOC, cloud cover and albedo) are described in Section 2.3. Further on, Section 2.4 states the methods used for data processing and analyses, including the ANN model and its validation, which is further described in Appendix A. Section 3 gives the results of this study and also compares them in a wider context, as Section 3.1 focuses on the climatology and changes

of spectral UV irradiance and the selected explanatory variables, and Section 3.2 presents the effects of the explanatory variables on the spectral UV irradiance as computed through the ANN model.

## 2 Materials and Methods

### 2.1 Study Site

The measurements assessed in this study took place at Marambio Base (S 64.14°, E 56.37°, 196 m a. s. l.). This permanent

Argentinean base was founded in 1969 and it is located on the ice-free Seymour Island in the Antarctic Peninsula region. The mean temperature does not exceed 0°C in any month, so snow cover, which increases UV radiation reflection, may be present at any time of the year. The cloudiest season is summer, with fog being prevalent especially in December and January (Lakkala et al., 2018; 2020). Marambio Base lies close to the edge of the southern polar vortex (Fig. 1), so during its existence in Antarctic winter and spring, the station lies just inside (about 60 % of all cases), on the edge, or just outside the vortex (Karhu

et al., 2003; Pazmiño et al., 2005).

### 2.2 Solar UV Spectra

The solar UV spectra have been obtained by the B199 double monochromator Brewer spectrophotometer over the period from February 2010 to January 2020. During this time, the B199 spectrophotometer was operated by the Czech Hydrometeorological Institute (CHMI) and the National Meteorological Service of Argentina (NMSI). The instrument was serviced each year in

January or February by specialists from CHMI and International Ozone Services, Inc., Canada. Each year, a calibration was performed using three to five traveling 50W lamps, which were calibrated right before the departure to Marambio Base using the B184 Brewer spectrophotometer in the CHMI Solar and Ozone Observatory in Hradec Králové, and three 1000W lamps S1450, S1451, and S1542 calibrated in the World Radiation Center in Davos. Moreover, in 2012 and 2016, the B199 spectrophotometer was calibrated against the world traveling standard B017 and in the same years, dispersion tests based on

wavelength shifts with the use of mercury and cadmium spectral lamps were carried out directly at Marambio, and the UV response file was updated. The maximal wavelengths shifts in the Brewer spectrum range (290-363 nm) were from +0.02 to -0.08 nm before the application of the new constants in 2018. A final calibration was performed in 2020 both at Marambio and after the instrument returned to Hradec Králové. The yearly calibration results, which were taken in account stepwise, yielded a maximum difference of -7 % in 2014. The mean absolute annual difference in the 2010–2020 period was 4.1 %. Generally,

the uncertainties of double-monochromator Brewer spectral measurements are up to 5 % (e.g., Lakkala et al., 2008). The instrument stability was further checked using the standard lamp ratios R6 and R5, which are related to the amounts of TOC and $SO_2$. The same ratios of intensities calculated from the halogen lamp show a long term stability of the instrument and the differences of extraterrestrial constant. Moreover, dead time test of photomultiplier and run-stop test of the slit mask contributed as the usual daily tests to check the functioning of the spectrophotometer. All this information was saved in the instrument checklist at the Solar and Ozone Observatory in Hradec Králové.

The B199 spectrophotometer obtains the spectral UV irradiance within the interval 286.5–363.0 nm, where the sampling interval is 0.5 nm and the full width half maximum (FWHM, i.e. the measured interval) is 0.6 nm. However, the very short wavelengths (below 300 nm) present higher uncertainty due to pronounced atmospheric attenuation and the sensitivity of the instrument itself (Bais, 1997; Bojkov et al., 1995). Moreover, the shortest wavelengths are also strongly absorbed by ozone, which leads to a very weak spectral UV irradiance at the Earth surface, which is affected by noise and the dark signal of the detector, and an uncertainty exceeding 10 % (Cordero et al., 2008; 2016). Therefore, only the wavelengths longer than 300 nm were used in this study.

Over the period of its operation, the B199 spectrophotometer obtained 45 073 solar UV spectra, each of which was quality-checked for spikes (Meinander et al., 2003), and the wavelength shift was analysed based on Fraunhofer lines using the SHICrivm software package (https://www.rivm.nl/en/uv-ozone-layer-and-climate/shicrivm). The subset of 23 260 spectra that passed quality control and was successfully paired with explanatory variables based on selected criteria (see Section 2.3), was used for this study. The temporal distribution of the spectra is shown in Fig. 2a. It can be seen that there were several data gaps, the longest of which occurred due to a remotely discovered technical failure of the instrument at the end of 2016. This mechanical problem was resolved early in 2017 during the regular maintenance so the measurements could continue. There were between one and 29 measurements per day (Fig. 2b), which were approximately evenly distributed over the course of day in summer, and centred around noon in spring and fall (Fig. 2c). The measurements took place at various SZAs, when in small-SZA months, the modus was near the small SZAs, but in large-SZA months, the modal category was shifted toward the larger SZAs (Fig. 2d).

## 2.3 Explanatory Variables

Each of the solar UV spectra used in this study was paired with the following explanatory variables: solar zenith angle (SZA), measured during each of the B199 spectral observations, total ozone column (TOC), cloud cover, and albedo. According to Petkov et al. (2016), these are to the most important variables affecting spectral UV irradiance at the nearby Mendel Station. The TOC values used in this study have also been acquired by the B199 Brewer spectrophotometer. Due to their very high precision, which can reach up to 0.15 % (Scarnato et al., 2010), only the Direct Sun measurements have been taken in account (as the Marambio weather is changeable, it was possible to make Direct Sun observations frequently). The solar UV spectra have been paired with these independent TOC observations, and only the TOC values within 60 minutes from the spectral measurements have been used. Therefore, several solar spectra were sometimes matched with the same TOC measurement,

provided the ozone observation was taken within 60 minutes of the spectral measurement and there was no other ozone observation with a shorter interval from the spectral measurement. When there was no match of the spectrum with a Direct Sun TOC measurement within the specified time interval, the spectrum was removed from the dataset. The 60 minute interval was chosen as a compromise between retaining a large solar UV spectra dataset and precision of the match with TOC. Due to the rapidly changing position of the polar vortex, the TOC values can vary greatly even within a relatively short time period. For example, a case study from the south Argentinean city Rio Gallegos by Orte et al. (2019) showed a drop in TOC by 70 DU in 24 hours, and Nichol and Valenti (1993) state that at South Pole, TOC may vary by more than 20 DU within several hours.

In order to express the effect of cloud cover, which was for the purpose of this study defined as the portion of sky covered in clouds, the ERA5 reanalysis data (Hersbach et al., 2020) were used. The spatial resolution was $0.25° \times 0.25°$, and the temporal resolution was 1 hour. This dataset, expressing the cloud cover in percentages, was chosen due to the best correlation (r = -0.26) with the Marambio Base cloud modification factor (CMF, Park et al., 2017), compared to other studied datasets (OMI cloud fraction and MERRA 2). As no direct cloud cover observations from Marambio were available for the use over the entire 2010–2020 study period, the ERA5 dataset was used despite its limitations (relatively large pixel size, no cloud type or cloud optical depth information, and therefore a relatively low correlation with CMF). CMF itself was calculated as the ratio between observed and clear-sky spectral irradiance (e.g., Lindfors et al., 2007) for each wavelength. In order to calculate a single CMF value for the entire spectrum, a weighted mean was used, so the CMFs at each wavelength were multiplied by the corresponding modelled clear-sky UV irradiances, summed up and divided by the clear-sky irradiance integrated through all studied wavelengths. CMF was determined from the ground based spectral UV irradiance data from B199, and the theoretical clear-sky spectral UV irradiance was estimated using the one dimensional DISORT solver of the libRadtran radiative transfer package (Mayer and Kylling, 2005). The input parameters in libRadtran were the day of year, SZA, B199 TOC, and albedo climatology (see further below). CMF itself could not be paired with the UV spectra, as it was not independent of the studied variable.

The climatological albedo values for the Marambio Base location have been taken from the OMI surface UV algorithm (OMUVB) product of the NASA's Aura Ozone Monitoring Instrument, which provides surface albedo climatology at 360 nm (Tanskanen et al., 2006), with the spatial resolution is 13 × 24 km at nadir. This dataset has only been used as an input to the spectral UV irradiance model described in sect. 2.4.

**2.4 Data Processing and Analysis**

The data analyses were carried out for the solar spectra, as well as for the explanatory variables (SZA, TOC, cloud cover), taking into account only the cases in which all the studied variables (UV spectrum, SZA TOC, cloud cover, and albedo climatology) were successfully paired (Section 2.2). At first, basic statistical characteristics were calculated for the entire period, individual months and seasons. Due to the non-normal distribution of the measurements and datasets (see e.g., Fig. 2d), median was chosen as the preferred measure of central tendency, alongside with the use of nonparametric testing where

applicable (e.g., Mann-Whitney U test and Kruskal-Wallis ANOVA, described in Kruskal and Wallis, 1952). All tests were performed at the significance level $\alpha = 0.05$.

In order to assess the change of spectral UV irradiance over the study period, daily median spectra have been calculated, as well as their relative differences to the 2010–2020 median. The solar irradiances at 305 and 340 nm have been further emphasized. These wavelengths have been chosen because of their different absorption by ozone (Bernhard and Stierle, 2020), but also because the results could be easily compared with other studies (e.g., Bernhard et al., 2005, 2006; Diaz et al., 2006; or Stamnes et al., 1991). At these wavelengths, the distribution of very high values was studied as well. A very high value was defined as belonging in the top 10 % of all measured irradiance values at the given wavelength.

The climatological effects of explanatory variables (SZA, TOC, and cloud cover) were assessed using an artificial dataset built by an ANN model (see further below). Such model was preferred to a parametric multiple regression model (e.g., Antón et al., 2005) because according to DeVaux et al. (1993), ANN models are suitable for noisy or intercorrelated (Table 1) input data and to express nonlinear relationships. Moreover, no a-priori assumption on the dependency shape was needed. Another advantage of ANN models is their ability to use even relatively simple datasets, which can make them desirable when not all variables needed for the use of a radiative transfer model (e.g., the precise structure of cloud cover) are available. In solar radiation climatology, ANN models are quite commonly used for various purposes, such as UV radiation forecasting or reconstructions (e.g., Barbero et al., 2006; Feister et al., 2008; Latosińska et al., 2015; Raksasat et al., 2021). In many cases, ANN models performed even better than conventional methods (Yadav and Chandel, 2014). However, the downside of ANN modelling is the creation of artifacts that can't be physically explained, and which will be pointed out in the Results section.

First of all, a large number of crossvalidated tests of randomly initialized neural networks with a random division of the dataset to training (70 % of data), testing (15 %) and validation (15 %) subsets were carried out. The training subset is used to find the quasi-optimal parameters of the neural network, while the testing subjects prevents overtraining (when it occurs, the error typically decreases on the training subset but increases on the testing one). A validation subset is used to independently assess the performance of the final trained network. The aim was to find the best set of predictors (i.e. SZA, TOC, cloud cover, and albedo climatology) and to establish the optimal complexity of the neural network, which would correspond to the complexity of underlying relations between the chosen predictors and spectral UV irradiance at individual wavelengths. To avoid both over- and under-parametrisation of the network, 22 neurons in the hidden layer provided the best results.

After this, ten different ANN multilayer perceptron models with logistic activation functions and 22 neurons in the hidden layer were built (TIBCO, 2023), each with four inputs (SZA, TOC, cloud cover, and albedo) and 127 outputs (spectral UV irradiance at the studied wavelengths in the interval 300.0–363.0 nm, with 0.5 nm resolution). Just like in the testing phase, each time, the dataset was randomly divided into a training (70 %), testing (15 %), and validation subset (15 %), whose statistical characteristics are shown in Appendix A. At the beginning of the training, each network was randomly initialized with a normal distribution with mean = 0 and standard deviation = 1. The stopping condition was set to the error improvement lower than 0.0000001 in the window of 200 cycles, and the error function was defined as the sum of squares.

Using the validation dataset, the performance of all ten models was assessed (see Appendix A), and the best one was selected based on the absolute and relative bias, root mean square error (RMSE), and correlation with the original data. According to the sensitivity analysis, SZA was the most important variable in the selected model, with the weight of 4.54, followed by cloud cover (1.14), TOC (1.05), and surface albedo (1.00).

However, even for the best of the ten models, a systematic bias was observed. So a correction was carried out: the median bias between measurement and model values was calculated from the entire dataset separately for each wavelength and month, and were subtracted from the model output. The correction for medians removed the systematic underestimation of spectral UV irradiance at almost all wavelengths (Fig. 3a,b). After the correction, 80 % of the data agreed to within $\pm80$ mW$\cdot$m$^{-2}\cdot$nm$^{-1}$. The relative differences were greater at very short wavelengths, but for wavelengths longer than approximately 310 nm, 80 % of the measured and modelled data agreed within roughly $\pm25$ %, which can also be seen in Fig. 3c. The correction improved the amount of data within $\pm1$, 5, 10, and 25 %, on average by 0.5, 1.6, 2.0, and 1.5 %, respectively. The coefficient of determination (R-squared, e.g. McClave and Dietrich, 1991) varied between 0.76 and 0.92 and it was greater at shorter wavelengths. The correction for medians increased the R-squared, on average by 0.1 %. RMSE was greater at longer wavelengths, and after the correction, it changed only within approximately $\pm0.1$ mW$\cdot$m$^{-2}\cdot$nm$^{-1}$.

After the best model was selected and the correction was performed, the effect of explanatory variables on the UV spectra was estimated. Of the four explanatory variables, one was selected and retained at its original value while the three other variables were fixed to their monthly medians (Table 2). The procedure repeated for each of the variables except albedo, whose effects were not studied further as its climatology was used. The use of individual months' medians was applied because neural networks generally present good results within the scope of the dataset, but they are not ideal for extrapolation, as they tend to overfit to the training dataset (e.g., Barbero et al., 2006; Raksasat et al., 2021). So, modelling spectral UV irradiance outside the dataset range, e.g., at a small SZA and very low TOC, could lead to very imprecise results. The breakdown of the dataset into individual months and the use of monthly medians prevented these inaccuracies, so it was possible to calculate the spectral UV irradiance change attributed to each of the four studied variables in the nine individual months in which B199 measurements were available (i.e. August to April).

Based on the above-described modelled data, the linearity and strength of the relationships were studied. At first, the relationships for each month at 305 and 340 nm were plotted, and then the mean absolute and relative change in spectral UV irradiance induced by the explanatory variables was assessed. The final part of the analysis consisted of several case studies, where the modelled values were compared to the actual observed spectra, with the aim to compare the effect of one variable with the effect of all studied parameters. Each time, two spectra, one with a very high and the other with a very low value of a particular explanatory variable, were chosen, and the observations were compared with the modelled spectrum, where all variables except one were fixed to their monthly medians. Due to the difference of monthly median and the actual values, the dissimilarities between the model and the observation showed also the combined effect of the other explanatory variables.

# 3 Results and Discussion

## 3.1 Solar UV spectra climatology and changes

Median solar spectral UV irradiance generally increases with increasing wavelength, but local minima exist at Fraunhofer lines and at wavelengths strongly absorbed in the Earth's atmosphere. Between 305 and 340 nm, the median spectral UV irradiance increases approximately 25 times in summer and over 100 times in early spring and late fall. The difference is likely given by the effect of SZA, when in the large-SZA months, the solar UV ray path through the ozone layer is longer. The increase is faster at short wavelengths up to ~330 nm (Fig. 4a), where it makes on average 7 mW·m$^{-2}$·nm$^{-1}$ but the shape is approximately exponential. At longer wavelengths, the increase changes to quasi-linear and slows down to 2 mW·m$^{-2}$·nm$^{-1}$. This behavior is well represented in all months but due to the respective position of the Sun at Marambio Base, it is steeper in small-SZA months (Fig. 4b). This happens, again, because at small SZA, the atmospheric path is shorter, so less UV radiation is absorbed by ozone and other atmospheric gases. At shorter wavelengths (up to approx. 330 nm), the increase in median spectral irradiance per nm varies from 3 mW·m$^{-2}$·nm$^{-1}$ in April to 10 mW·m$^{-2}$·nm$^{-1}$ in December and January. At longer wavelengths, the spectral irradiance variability increase is smaller and makes 1 mW·m$^{-2}$·nm$^{-1}$ in August, September, March and April, and 2 mW·m$^{-2}$·nm$^{-1}$ in other months. The shape of the spectrum, given by the extraterrestrial irradiance and atmospheric attenuation, remains similar in all months, with a higher variability between local maxima and minima in spring and autumn (effect of small SZA).

The relative differences of monthly medians to the overall median spectrum, which takes in account all observations from the 2010–2020 period, vary especially at wavelengths to about 320 nm (Fig. 4c). At 300 nm, the UV irradiance exceeds the overall median by up to 139 % in January and reaches only 7 % of it in April. However, at 363 nm, the monthly medians make up between 135 and 52 % of the overall one, so the difference is smaller. The smaller relative differences at short wavelengths (approximately 300–305 nm) in September and larger ones in October may be possibly attributed to the effect of variable ozone amount, as the ozone deficiency causes a relative increase in UV irradiance at short wavelengths. The wave-like structure in the Huggins band, visible in certain months, is likely also conditioned by the amount of ozone relative to the general central tendency (e.g., Gorshelev et al., 2014).

TOC (Fig. 4d) exhibits a pattern typical for the Antarctic environment, i.e. depletion in austral spring, explained for example in the WMO ozone assessment (WMO, 2022). In September, the median TOC value drops below the 220 DU level, and the absolute minimum, 113.7 DU, was recorded on 3 November 2015. In November and December, the median TOC values grow above the summer medians, which is likely due to the advection of subpolar, ozone rich air masses into the area of the decaying polar vortex. Over the study period, the highest TOC value was recorded on 6 November 2011, when it reached 402.0 DU.

The yearly cycle, as well as the short- and long-term fluctuations of TOC in the coastal Antarctic region depend both on chemical and dynamical influences, of which the chemical ones (like the catalytic reactions with the contribution of man-made chemicals) are now quite well understood (e.g., Solomon, 1999). The dynamical influences include the Brewer-Dobson circulation, which causes the poleward transport of ozone from the tropic to the poles, and subsequent accumulation of ozone

in the polar regions in winter, as no UV radiation is present to induce ozone loss (Weber et al., 2011). Ozone depletion through catalytic reactions starts in early spring and low TOC values are present till the breakdown of the polar vortex, which is caused by the dynamical effect of planetary waves and has much year-to-year variability (e.g., Shepherd, 2008), so it was possible to observe both the absolute ozone minimum and maximum in one month (November), only in different years. Moreover, the main reason why both TOC minima and maxima were reached in November is the position of Marambio near the edge of polar vortex, so the station can be either inside it and exhibiting extremely low TOC amounts, or outside, affected by the subpolar air masses, which are typically rich in ozone in November (e.g., Diaz et al., 2006). The variability of TOC, which is much larger in spring than in any other season, is well visible in Fig. 4d.

The large variability of cloud cover at Marambio Base, with the median value of 68 % and mean of 59 %, is expressed in Fig. 4e. For the latitude of Marambio, Lachlan-Cope (2010), who used latitudinal means obtained from satellite observations, presents a higher mean cloud cover (around 80 %). This difference is due to the position of Marambio at the eastern, i.e. leeward, and therefore less cloudy side of the Antarctic Peninsula. There is a hint of weak yearly cycle with maxima in September and minima in February, but the differences between the months are not statistically significant and therefore not conclusive. Based on satellite data, for the location of Marambio, all Adhikari et al. (2012), Lachlan-Cope (2010) and Scott et al. (2017) found the cloud cover minima in winter and maxima in summer. However, due to the large cloud cover variability, mentioned also in Aun et al. (2020), the results presented in this study might have been affected by the choice of study period, and remain statistically not significant.

According to the surface albedo climatology, mean albedo at Marambio Base fluctuates between 0.42 in January and 0.61 in August. Due to the snow cover melting and sea ice reduction, albedo decreases in the summer season, but in March the sea ice recovers and albedo increases again. However, albedo can occasionally exceed 0.8 even in summer, which is related to snowfall events, as at the Antarctic Peninsula, most precipitation falls in solid form (e.g., Ambrožová et al., 2020; Carrasco and Cordero, 2020; Engel et al., 2022).

The spectral UV irradiance represented in Fig. 5a shows a clear year to year variability with minima in early spring and late autumn and maxima in Antarctic summer. However, the attenuation of spectral UV irradiance in the atmosphere, caused likely by the action of clouds and atmospheric gases, frequently disrupts the dominant pattern. The lower intensity at some wavelengths (e.g., 316.5 nm or 358.5 nm), which is in Fig. 5a represented by the relatively darker vertical lines, has been caused by the presence of Fraunhofer absorption lines.

Although the absolute variability of spectral UV irradiance is highest at the longer wavelengths, the relative variability (Fig. 5b) is greatest at the shortest wavelengths, as seen from the presence of highest and lowest values of the ratio between the daily median spectra and the overall median (in Fig. 5b, where it is expressed on logarithmic scale). Also here, the minima occur in early spring and late autumn, but the maxima can be recorded both in summer and spring. This is further documented by Fig. 5c and 5e, which implies that at 305 nm, solar UV irradiance maxima can occur in October, as well as in December or January. For example, in the seasons 2010–2011 or 2017–2018 the summer (January) maximum, caused likely by low cloud cover, is more pronounced. However, in other seasons, such as in 2011–2012, 2014–2015 or 2018–2019, UV irradiance at 305

nm peaks in spring, particularly in October and November. The spring peak is affected mostly by ozone, which attenuates especially radiation of very short wavelengths (e.g., Bais et al., 1993; Gorshelev et al. 2014). Very short wavelengths seem to follow a similar pattern with a large, ozone-induced variability in spring at different locations in Antarctica, such as McMurdo Station (Bernhard et al., 2006), or Palmer Station (Bernhard et al., 2005), but also in the southernmost South America (Diaz et al., 2006). On the contrary, at 340 nm (Fig. 5d), the summer maximum is higher in all cases except the 2014–2015 season. This wavelength is considered largely unaffected by ozone, however, despite being affected much less than shorter wavelengths, in the cases of some deep ozone holes irradiance increases can be seen even at longer wavelengths. This happened for example in seasons 2013–2014, the ozone hole was short-lasting but deep and well correlated with a drop in cloud cover, or in 2018–2019, where a drop in cloud cover was also observed.

Another parameter, important for the UV radiation attenuation, is cloud cover. As seen from Table 2 and Fig. 4e, during the study period, maxima tend to occur in spring and minima in early fall, but there is much year to year variability (Fig. 5f). Near the Antarctic Peninsula, cloud cover can attenuate more than 90 % of UV irradiance, but in the case of a relatively thin cloud cover and a short optical path through the atmosphere, this number can shrink to as little as 30 % (Lee et al., 2015). In the case of very thin clouds (e.g., cirrus), the attenuation of UV radiation could reach only about 5 % (Kuchinke and Nunez, 1999). However, this amount is likely not the same at all wavelengths, as due to the increased scattering of shorter wavelengths in the atmosphere and thus the decreasing proportion of direct radiation (described e.g., in Schwander et al., 2002), clouds affect solar radiation at longer wavelengths more. Nevertheless, it seems that during cloudy days, such as in September 2010 or August to October 2017, spectral UV irradiances at Marambio drop at both short and long wavelengths, whereas below-median cloud cover (e.g., December 2015) caused high summer spectral UV irradiance peaks.

Surface properties, represented by albedo, are another important factor affecting the spectral UV irradiance. Snow or sea ice can reflect 50 to 95 % of UV radiation, depending on SZA and surface quality, such as soot content. However, the ocean without sea ice only reflects 5–20 % of UV radiation, which will lead to lower UV radiation compared to snow cover or glaciated areas (Cordero et al., 2014; Zhou et al., 2019). Around Marambio Base, albedo reaches maxima at the end of winter (August to October), and minima in late summer (January to February).

Very high spectral UV irradiances at Marambio Base at 305 and 340 nm, which were defined as the highest 10 % of all recorded values, were identified especially in spring and summer, typically from October to February (Fig. 6). Due to the effect of low ozone values, UV irradiance at 305 nm often peaks earlier than at 340 nm, and there are large differences between the number of high UV irradiances at 305 and 340 nm in each month. The two datasets only share less than 52 % of variability, which proves that it is important to study UV irradiance at separate wavelengths individually. For example, in October 2010, there were no high UV irradiances at 340 nm, but there were 34 of them at 305 nm. Although the ozone reduction in 2010 was relatively small (Fig. 5e, De Laat and Van Weele, 2011), relatively low cloud cover (median of 43 %, as seen in Fig. 5f) allowed the UV irradiance at 305 nm to reach very high values. Similarly, in the season 2011–2012, there were 12 cases of very high UV irradiances at 305 nm already in September and 84 of them in October, but there were only 3 cases at 340 nm in October 2011. In 2011, the ozone hole was relatively severe (Fig. 5e, Klekociuk et al., 2014), so it strongly affected the UV

irradiance at 305 nm. Another similar example, where relatively deep and stable polar vortex with low ozone content increased the number of very high irradiance measurements at 305 nm but not at 340 nm, was the season 2018–2019 (Fig. 5e, and e.g., Klekociuk et al., 2021). On the contrary, in November 2017, only 18 cases of high UV irradiances were recorded at 305 nm,
compared to 113 cases at 340 nm. This was due to very high TOC in November 2011 (the median value of TOC was 357.4 DU), which attenuated the solar irradiance at 305 nm.

## 3.2 Effects of explanatory variables on spectral UV irradiance

As seen from Fig. 7–9, the relationships between spectral UV irradiance and explanatory variables, as modelled by the ANN model described in Section 2.4, are often not linear, and vary based on wavelength. The relationships and amounts of change
for different months also vary, which demonstrates that they are affected by other variables. Fig. 7 shows the relationship between SZA and UV irradiance with the other explanatory variable fixed to their monthly medians. Similar relationships between TOC and UV irradiance and between cloud cover and UV irradiance are shown in Figs. 8 and 9, respectively.
Of the four studied variables, SZA is the most important one affecting the spectral UV irradiance at Marambio Base (Fig. 7). The relationship between UV irradiance and SZA at 305 nm (Fig. 7a) is not linear, as the increase in UV irradiance at lower
SZA is faster than at large SZA values. Compared to other months, UV irradiance at 305 nm in September and October increases faster with the decrease in SZA, which is likely caused by its enhancements by the low TOC values in the ozone-hole period. The relationship between SZA and the modelled spectral UV irradiance is getting more linear with increasing wavelength (Fig. 7b), and the relative differences between individual months are smaller. The nonlinearity of the relationship can be explained by the solar spectral UV irradiance being inversely proportional to the cosine of SZA, and by the increasing
absorption and scattering in the atmosphere as the optical path lengthens (e.g., Kerr and Fioletov, 2008). Although the absolute mean differences in spectral UV irradiance for the 1° increase in SZA are greater at wavelengths longer than about 320 nm, the relative changes are largest at the very short wavelengths to approximately 310 nm (Fig. 7c). For example, at 340 nm, a 1° increase in SZA causes between 3 % decline in UV irradiance in December and 8 % in August, but at 305 nm, this change varies between 6 % in December and 18 % in March, provided all other variables are fixed to their monthly medians. Webb
and Engelsen (2008) explain this by the increasing path of UV radiation through the atmospheric ozone layer, so the short wavelengths are attenuated more than the longer ones. The spectral dependence of the relationship between the spectral UV irradiance and SZA has also been shown in the case studies by Kerr and Fioletov (2008) or Tarasick et al. (2003), and indirectly also by the 340/305 nm ratio studied by Stamnes et al. (1991). However, Fig. 7 includes several ANN artifacts that need to be taken into account during the interpretation, such as the slow-down or even inversion of the spectral UV irradiance increase
toward the smallest SZA values (Fig. 7a, b), similarly to the UV irradiance presented in Fig. 7a, which does not asymptotically decrease to zero at large SZAs, or the fluctuations of relative UV irradiance change in large-SZA months at wavelengths shorter than approximately 310 nm (Fig. 7c).
The fact that ozone affects spectral UV irradiance more at shorter wavelengths can be seen from Fig. 8. UV irradiance at 305 nm increases with the decrease in TOC (Fig. 8a) at an average rate of up to 2 $mW \cdot m^{-2} \cdot nm^{-1}$ per 10 DU in November, but the

relationship is nonlinear, with a faster increase at low TOC values. A similar, approximately exponential relationship of TOC and UV irradiance at 300 nm from South Argentina has been presented by Bojkov et al. (1995). The differences in the relationships between the studied months are likely caused by the effect of SZA, which is expressed in Fig. 7 or for example in Kerr and Fioletov (2008). At 340 nm, TOC is only a weak absorber of UV radiation (e.g., Gorshelev et al., 2014), so it does not have a profound effect at this wavelength (Fig. 8b). The rather complex shape of the relationship is likely to be an artifact of the ANN model. The mean absolute change in spectral UV irradiance induced by a 10 DU increase in TOC is greatest at about 317.5 nm, while the relative change is greatest at the shortest wavelengths, and it is getting close to 0 at the longer ones (Fig. 8c; also described in Bais et al., 1993). The Huggins band wavy structure is clearly visible between approximately 315 and 335 nm. However, at large SZA (approximately 75 °), the shape of vertical ozone profile (e.g., Čížková et al., 2018) may play a substantial role in UV radiation absorption in the optical path, as different vertical distributions of ozone may lead to similar TOC values.

According to the ANN model, the effect of cloud cover (Fig. 9) seems to be generally quite weak at both 305 and 340 nm (Fig. 9a,b), being greater near the fully cloudy sky. The absolute change in spectral UV irradiance caused by a 10 % increase in cloud cover is greatest at the longer wavelengths, but the relative change is largest at the shortest wavelengths, especially in large-SZA months (Fig. 9c). The large fluctuations of the relative UV irradiance change in the wavelength interval of approximately 300–313 nm are likely to be an artifact of the ANN model. From about 313 nm onward, the relative change of modelled spectral UV irradiance remains almost the same, and it ranges from about -0.2 % in April to approximately -2.2 % in January. Although Bais et al. (1993) did not find any spectral dependence of the relationship between spectral UV irradiance and cloud cover, Bernhard et al. (2004), Seckmeyer et al. (1996) and Kylling at al. (1997) argue that due to spectrally dependent Rayleigh scattering of UV radiation at top of cloud layer, the cloud transmittance is higher at shorter wavelengths. Similarly, Schwander et al. (2002) found that in the UVA region, the attenuation of UV radiation by clouds increases with increasing wavelength. The clouds can attenuate up to 99 % of UV radiation, however, their effect is generally weaker over the Antarctic continent, possibly explaining the relatively small changes detected by the modelling experiment. This happens because the Antarctic clouds are thinner than the ones in middle and low latitudes, affecting the portion of UV radiation penetrating through them (Calbó et al., 2005; Lubin and Frederick, 1991). Bernard et al. (2006) reported that at McMurdo, clouds cause only 10 % reduction in UV irradiance at 345 nm on average. Also Lee et al. (2015) found a relatively weak effect of cloud cover, when in the Antarctic Peninsula region, the increase in cloud cover from clear sky to overcast caused only about 30 % decrease in erythemal UV irradiance, and this change was even smaller at large SZA. Similarly to this study, they showed a nonlinear relationship with a faster decline in erythemal UV irradiance when the cloud cover exceeds approximately 5 octas. Another possible reason for the relatively weak influence of clouds on UV irradiance at Marambio, especially in spring and fall, is the high albedo of snow-covered surfaces and sea ice, which reduces cloud effects due to multiple scattering between the clouds and highly reflective surface (Nichol et al., 2003).

In the last part of the study (Fig. 10), an assessment was carried out, looking at the effects of SZA, TOC, and cloud cover on the UV spectra on selected October days with extreme values of each parameter. The modelled results show the potential effect

of a single variable in its high and low extremes with all other parameters were fixed to their monthly medians. The real observations, however, point out to the effect of all variables combined, which caused a different UV irradiance change compared to the theoretical modelled values.

Looking at individual case studies featuring solar spectra measured at a large and a small SZA (Fig. 10a), the modelled theoretical effect confirms the relative change of spectral UV irradiance is greatest at the shortest wavelengths. However, due to the interplay of other variables, in the selected small-SZA case the potential UV irradiance increase at short wavelengths (approximately below 320 nm) was hindered by a very high ozone content (about 120 DU higher than the monthly median). Therefore, it shows that despite SZA is the most important of the studied variables affecting UV irradiance, its effect can be at least partly overrun by the actions of other explanatory variables.

The two cases from 28 October 2010 and 3 October 2011, presented in Fig. 10b, show the measurements with record-high and record-low October TOC, differing from each other by as much as 274 DU. It can be seen that the potential change is greatest at short wavelengths, whereas at longer ones, it is getting close to zero. However, in the low-TOC case (3 October 2011), much of the potential UV irradiance increase is masked by the effect of SZA, which was in reality more than 15° higher than the monthly median. At short wavelengths, the extremely low TOC only caused the difference to monthly median to stagnate, acting opposite to the effect of large SZA (presented in Fig. 10a). Therefore, even at low TOC conditions, spectral UV irradiance may not be record high due to other factors (e.g., SZA, cloud cover).

Fig. 10c represents the effect of cloud cover on 18 October 2010 and 3 October 2014. The potential effect of cloud cover is greater at longer wavelengths, where it reaches about 20–25 %. However, in the high cloud cover case (18 October 2010, fully clouded sky), the difference is even greater than the calculated potential, even when the SZA is only about than 1° higher than the monthly median and TOC is approximately the same. This can be conditioned by the presence of a specific cloud type, as thicker, low clouds attenuate more UV irradiance than mid or high ones. The relatively smaller UV irradiance increase at short wavelengths in the low cloud cover case (3 October 2014) may be caused by the effect of TOC, which was more than 10 DU higher than the monthly median.

## 4 Summary and Conclusion

UV radiation is an important factor affecting the life on Earth, but its effects are spectrally dependent. However, in the southern polar areas, a detailed analysis of solar UV spectra and their response to different atmospheric and terrestrial factors was still missing. In this study a comprehensive analysis of spectral UV radiation, based on 10 years of Brewer spectrophotometer measurements, was carried out using the approach of neural network modelling. It helped to confirm the strongly wavelength-dependent behaviour of spectral UV irradiance, showing the importance of spectral UV radiation assessment. The absolute differences between the spectral UV irradiance spectra generally increase with increasing wavelength, but the relative differences are greater at the shortest wavelengths. In the studied months (August to April) at Marambio Base, the highest median spectral UV irradiance was recorded in January (139 % of the overall median at 300 nm, 135 % at 363 nm), whereas

the lowest one in April (7 % of the overall median at 300 nm, 52 % at 363 nm). However, in some seasons (e.g., 2011–2012, 2014–2015, 2018–2019), spectral UV irradiance at the shortest studied wavelengths peaked in spring (October and November). The temporal distributions of very high UV irradiances at 305 and 340 nm only share less than 52 % of variability, showing that UV irradiances at different parts of the spectrum respond differently to the driving factors, such as SZA, TOC, or cloud
cover.

SZA is the most important variable affecting spectral UV irradiance at Marambio Base. According to the ANN model, the increase in spectral UV irradiance with the decrease in SZA is faster at short wavelengths. At 305 nm, a 1° decline in SZA equals on average between 6 and 18 % increase in UV irradiance, but only between 3 and 8 % at 340 nm, depending on the month. At Marambio Base, TOC shows a pattern typical for the southern polar conditions with a pronounced depletion in
spring. Over the 2010–2020 period, the spring depletion was variable, with deep, long-lasting ozone holes in the seasons 2011–2012 or 2015–2016. The absolute TOC minimum was reached on 3 November 2015. On the other hand, the 2019 ozone hole was the smallest on record. In seasons with a deep, long-lasting ozone hole, the spring peak of the spectral UV irradiances is most visible at very short wavelengths. The relationship between TOC and UV irradiance at 305 nm is non-linear, with a faster increase at low TOC values. At this wavelength, a 10 DU increase in TOC causes a decrease in UV radiation, which totals,
depending on the month, on average between 7 and 13 %. At 340 nm, TOC does not have a profound effect.

Due to the relatively thin Antarctic clouds, the effect of cloud cover on spectral UV irradiance was found rather weak. The increase in cloud cover by 10 % caused between 1–2 % decrease in UV irradiance at both 305 and 340 nm, and the attenuation increased fastest between 90 % and 100 % cloud cover.

Additional efforts are needed to extend this study by the evaluation of other variables, such as other atmospheric gases
(tropospheric ozone or sulphur dioxide), the shape of vertical ozone profiles, but especially in-situ measured albedo, snow cover occurrence or the sea ice extent in a given perimeter around Marambio Base, because the albedo of different surfaces, including snow, is spectrally dependent. Similarly, cloud type may also be studied, as it is also an important variable affecting spectral UV irradiance. It would be also beneficial to intercompare radiative transfer, ANN, and regression spectral UV radiation models, provided all necessary input variables are available. Lastly, it is important to note that the results of this
study might have been influenced by the peculiar characteristics of the given location, so they cannot be easily applied to the entire Antarctic Peninsula region.

**Appendix A**

**Artificial Neural Network model development and validation**

Of the ten ANN models we built, nine (ANN02 to ANN10) behaved in a similar way, while one (ANN01) was different, and
showing the best performance confirmed by validation statistics. The differences between the models did not result from the ANN setting, which remained the same, but occurred due to the random initialization of the models and the random split of the dataset to training (70 %), testing (15 %), and validation (15 %) subsets. However, according to Fig. A1, the statistical

characteristics of the subsets were similar in the case of all ten models and no major differences were observed. As seen from Fig. A2, the model ANN01 had the most data within ± 5, respective ± 10 % from observations, and it had the largest R-squared and lowest RMSE of all ten models. However, the model was biased toward underestimation of spectral UV irradiance throughout most of the spectrum. For the purpose of the study, it was best to choose a model with the best precision, i.e. the lowest variability of results, highest R-squared and lowest RMSE (model ANN01). Also, it was possible to tackle the bias present within the model using a simple median correction described in section 2.4.

**Data availability**

The solar radiation data used for the study are property of the Czech Hydrometeorological Institute, Hradec Králové, Czech Republic and are the subject of the data policy of the above-mentioned institution. Any person interested in the underlying data should contact Ladislav Metelka, the head of the Solar and Ozone Observatory of the Czech Hydrometeorological Institute, Hradec Králové (email: ladislav.metelka@chmi.cz). The satellite and reanalyzed data courtesy of NASA and ECMWF, respectively.

**Author contribution**

K.Č., K.L. and L.M. designed the study, L.M. and M.S. provided the resources, L.M. and M.S. designed the ANN model and K.Č. validated it, K.Č. performed the data analyses and prepared the original manuscript draft, and K.L. and L.M. reviewed and edited it.

**Competing interests**

The authors declare that they have no conflict of interest.

**Special issue statement**

The work was presented online at the Quadrennial Ozone Symposium 2021 (poster No. SAT3_9) and we have expressed the interest to submit the paper to the joint Special Issue "Atmospheric ozone and related species in the early 2020s: latest results and trends" in ACP/AMT journals following on the QOS2021.

**Acknowledgments**

This research was supported by the State Environmental Fund of the Czech Republic, project of CHMI no. 03461022 'Monitoring of the ozone layer and UV radiation in Antarctica'; by the Czech Antarctic Research Programme 2022 (VAN

2022), funded by the Ministry of Education, Youth and Sports of the Czech Republic; and the project of Masaryk University (MUNI/A/1393/2021). The authors would like to thank the European Centre for Medium-Range Weather Forecasts for providing the ERA5 cloud cover data product and NASA AURA Validation Data Center for the surface albedo climatology retrieved from OMI measurements. The authors are also grateful to the LAMBI laboratory and Marambio Base staff for operating the B199 Brewer spectrophotometer, namely to Janouch, M., Sieger, L., Brohart, M., Stráník, M., Hrabčák, P., Savastiouk, V, and Ochoa, H. The authors also express thanks to all anonymous referees for their constructive criticism and manuscript improvement suggestions.

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

**Table 1: Intercorrelation of explanatory variables (SZA = solar zenith angle; TOC = total ozone column; CLD = cloud cover; ALB = albedo). Statistically significant results (α = 0.05) are marked with an asterisk.**

|     | SZA    | TOC    | CLD   |
| --- | ------ | ------ | ----- |
| TOC | -0.23* | ·      | ·     |
| CLD | -0.04* | -0.12* | ·     |
| ALB | 0.20*  | -0.19* | -0.01 |

**Table 2: Monthly median values of explanatory variables (SZA = solar zenith angle; TOC = total ozone column; CLD = cloud cover; ALB = albedo) at Marambio Base in 2010–2020, which were used to fix the variables in the ANN model runs.**

|     | AUG   | SEP   | OCT   | NOV   | DEC   | JAN   | FEB   | MAR   | APR   |
|-----|-------|-------|-------|-------|-------|-------|-------|-------|-------|
| SZA | 76.11 | 70.13 | 61.75 | 55.60 | 52.12 | 53.50 | 59.27 | 67.12 | 74.59 |
| TOC | 246.5 | 205.5 | 223.7 | 300.2 | 311.4 | 287.8 | 283.7 | 273.6 | 283.0 |
| CLD | 67    | 76    | 73    | 70    | 66    | 64    | 62    | 67    | 71    |
| ALB | 0.61  | 0.60  | 0.62  | 0.57  | 0.47  | 0.42  | 0.43  | 0.51  | 0.59  |

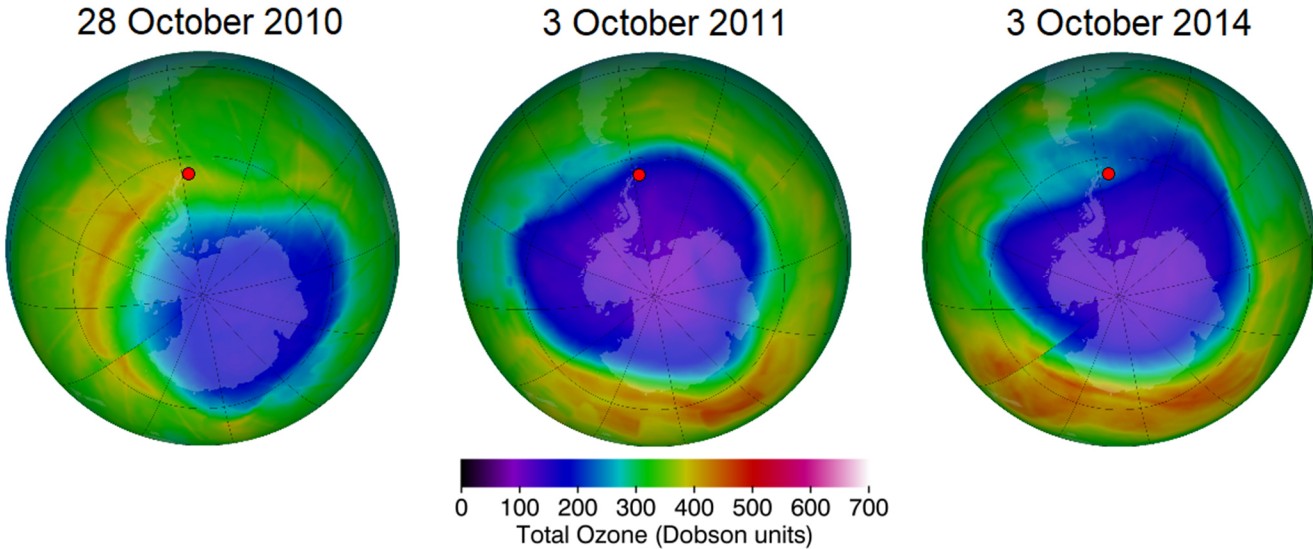

**Figure 1: The approximate position of Marambio Base (red dot) relative to the southern polar vortex (outside, inside, and on the edge) on different October days. Image adapted from Ozone Hole Watch (2022).**

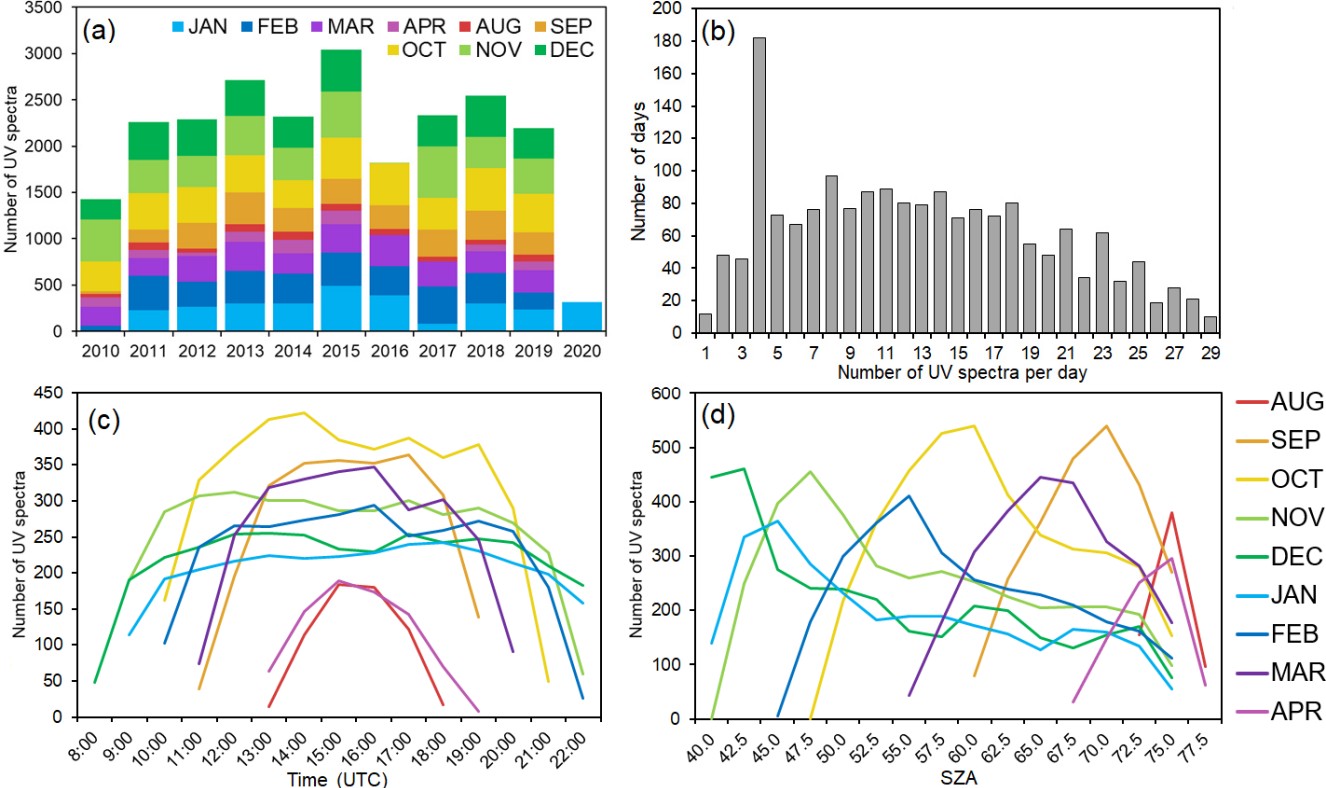

**Figure 2: Number of solar UV spectra used in this study measured by the B199 Brewer spectrophotometer at Marambio Base over**
**the period 2010–2020, where (a) represents the distribution over the years, (b) is the number of spectral UV measurements per day, (c) shows the distribution of measurements in individual months based on the time of day, and (d) is the measurement distribution in individual months based on SZA.**

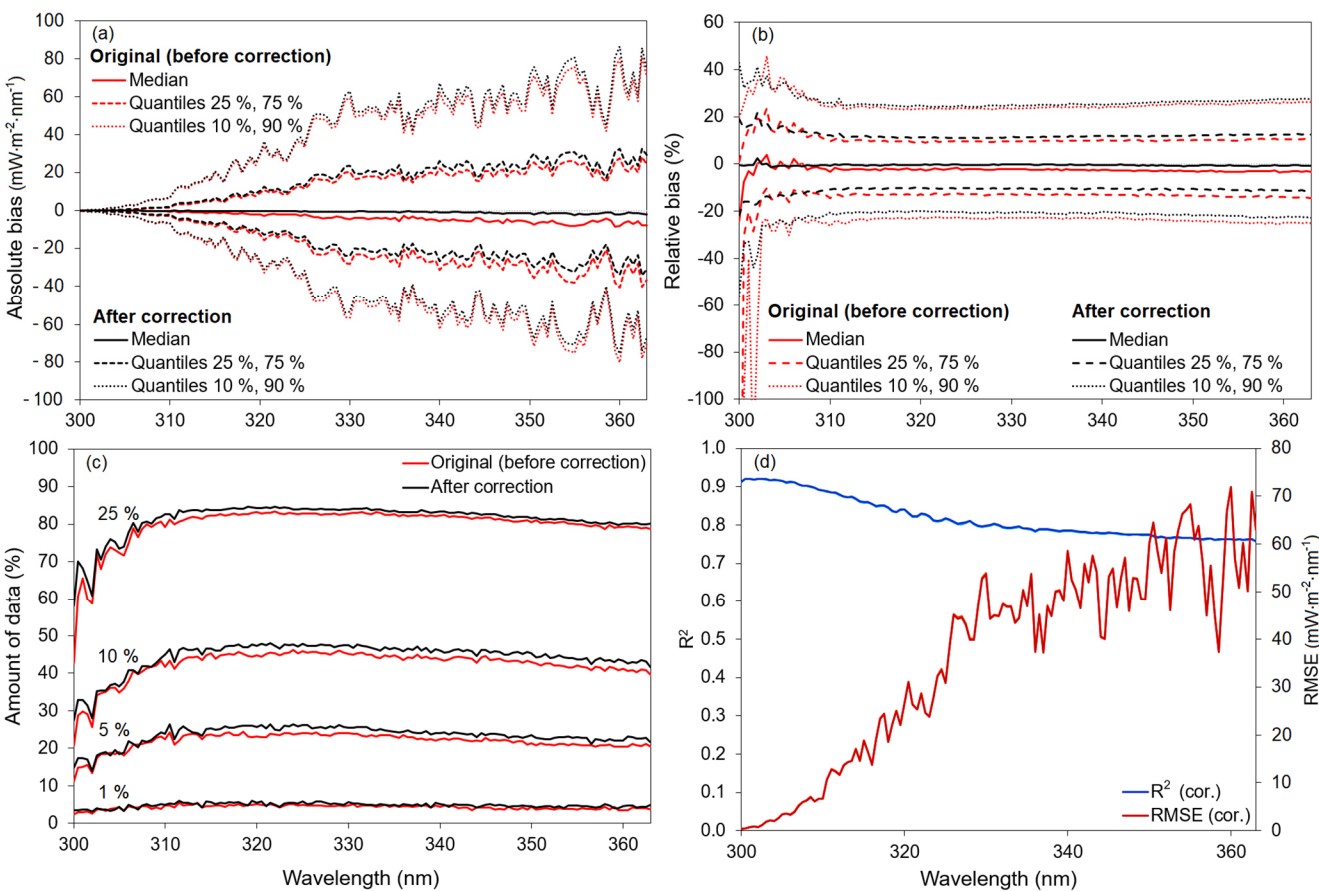

Figure 3: Validation of the ANN model before and after the median correction, where (a) represents the absolute bias quantiles, (b) are the relative bias quantiles, (c) is the amount of data within 1, 5, 10, and 25 % from observations, and (d) is the R-squared and RMSE.


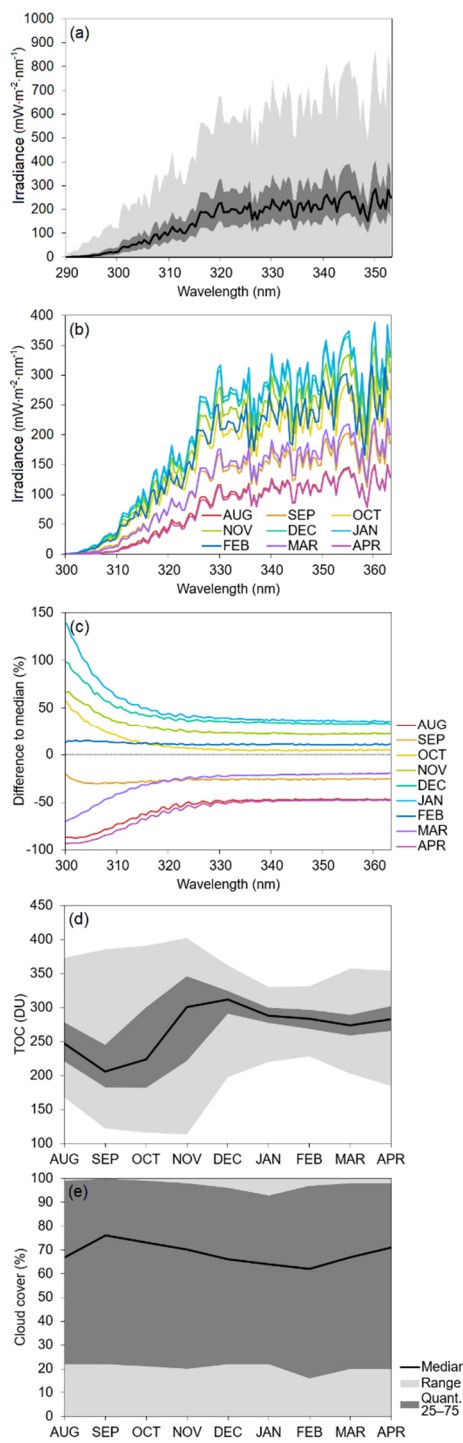

**Figure 4: Observed solar UV spectra and variables related to them obtained at Marambio Base in 2010–2020, where (a) represents statistical characteristics of all spectra, (b) monthly medians, and (c) relative differences of monthly medians to overall median, and (d), and (e) show the climatology of TOC, and cloud cover, respectively.**

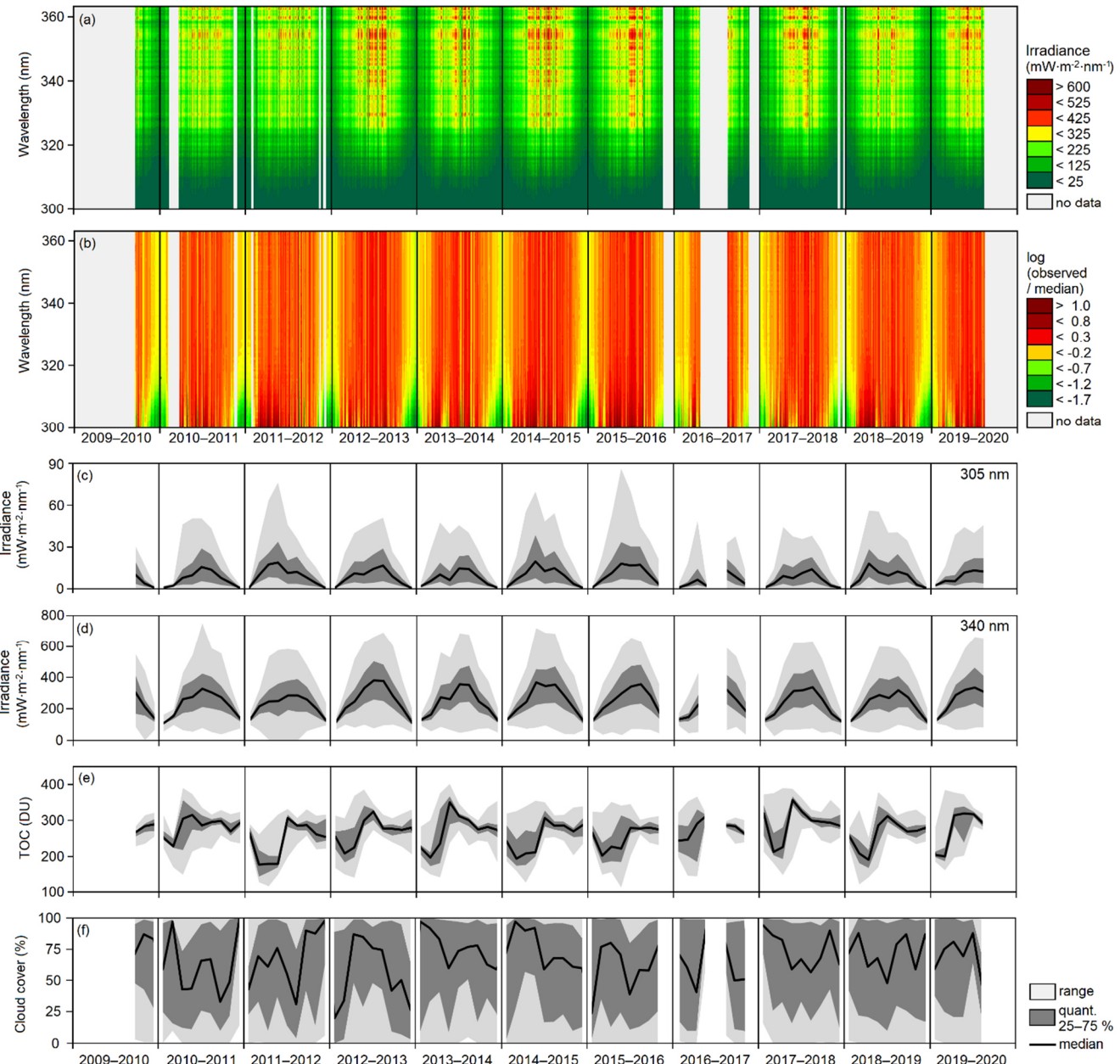

**Figure 5: Spectral UV irradiance and related variables at Marambio Base over the period 2010–2020, where (a) represents the daily median intensities of spectral UV irradiance, (b) the daily median spectral UV irradiances relative to the overall median, on logarithmic scale, at a given wavelength, (c) the monthly variability of solar irradiance at 305 nm, (d) the monthly variability of solar irradiance at 340 nm, (e) the monthly variability of TOC, and (f) the monthly variability of cloud cover.**


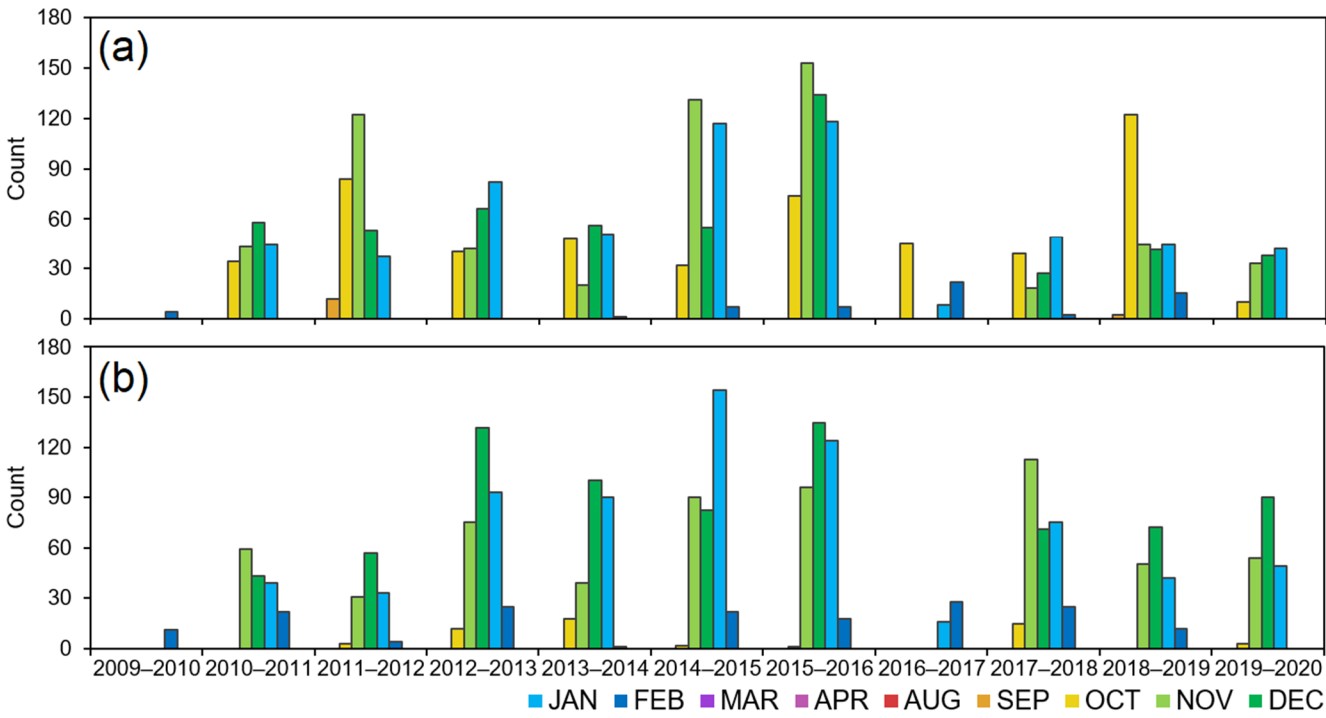

**Figure 6: Monthly distribution of the 10 % highest UV irradiances at (a) 305 nm, and (b) 340 nm, at Marambio Base, 2010–2020.**


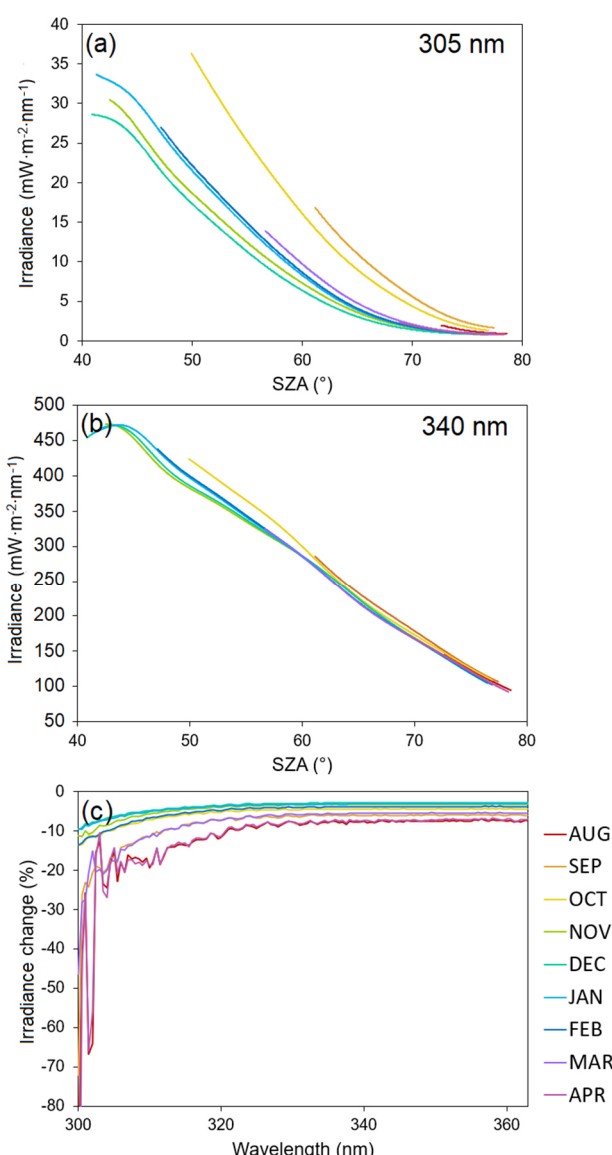

**Figure 7: Modelled relationships between UV irradiance and SZA for different months. Panel (a) and (b) show the modelled UV irradiance at 305 nm (a) and 340 nm (b) as a function of SZA. Panel (c) shows the relative change in spectral UV irradiance resulting from an increase in SZA by 1°, calculated with reference to the monthly median spectra.**


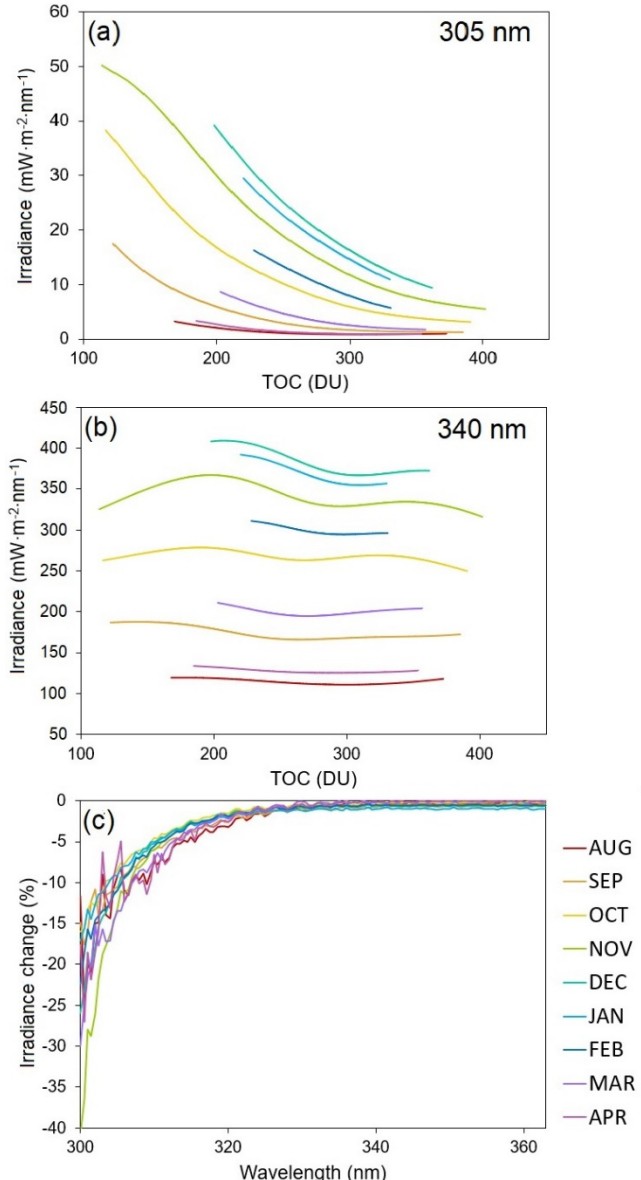

**Figure 8: Modelled relationships between UV irradiance and TOC for different months. Panel (a) and (b) show the modelled UV irradiance at 305 nm (a) and 340 nm (b) as a function of TOC. Panel (c) shows the relative change in spectral UV irradiance resulting from an increase in TOC by 10 DU, calculated with reference to the monthly median spectra.**

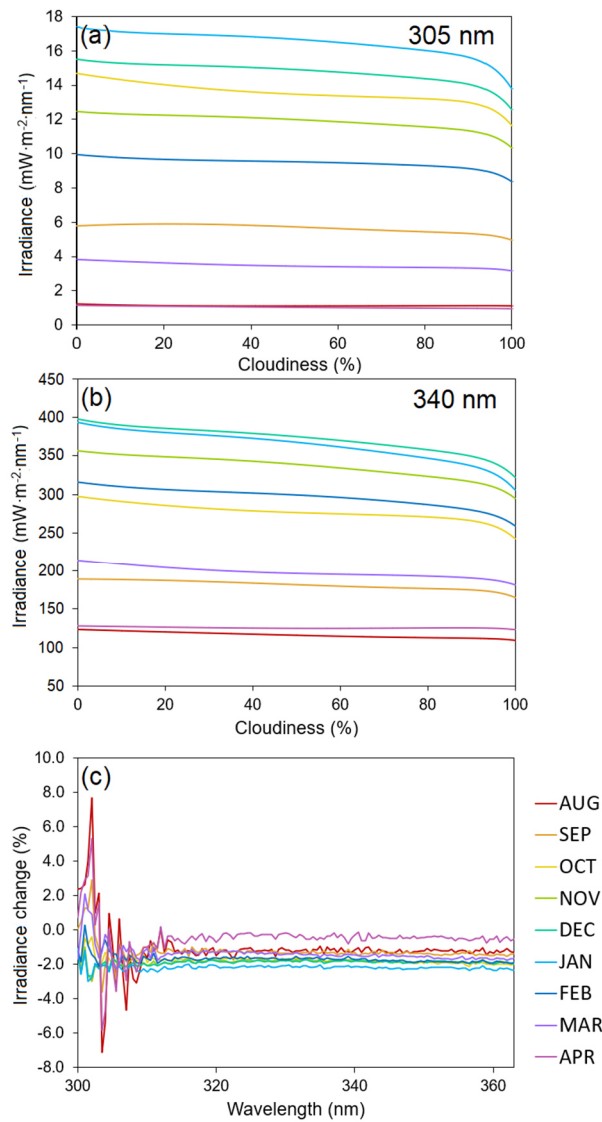


**Figure 9: Modelled relationships between UV irradiance and cloud cover for different months. Panel (a) and (b) show the modelled UV irradiance at 305 nm (a) and 340 nm (b) as a function of cloud cover. Panel (c) shows the relative change in spectral UV irradiance resulting from an increase in cloud cover by 10 %, calculated with reference to the monthly median spectra.**

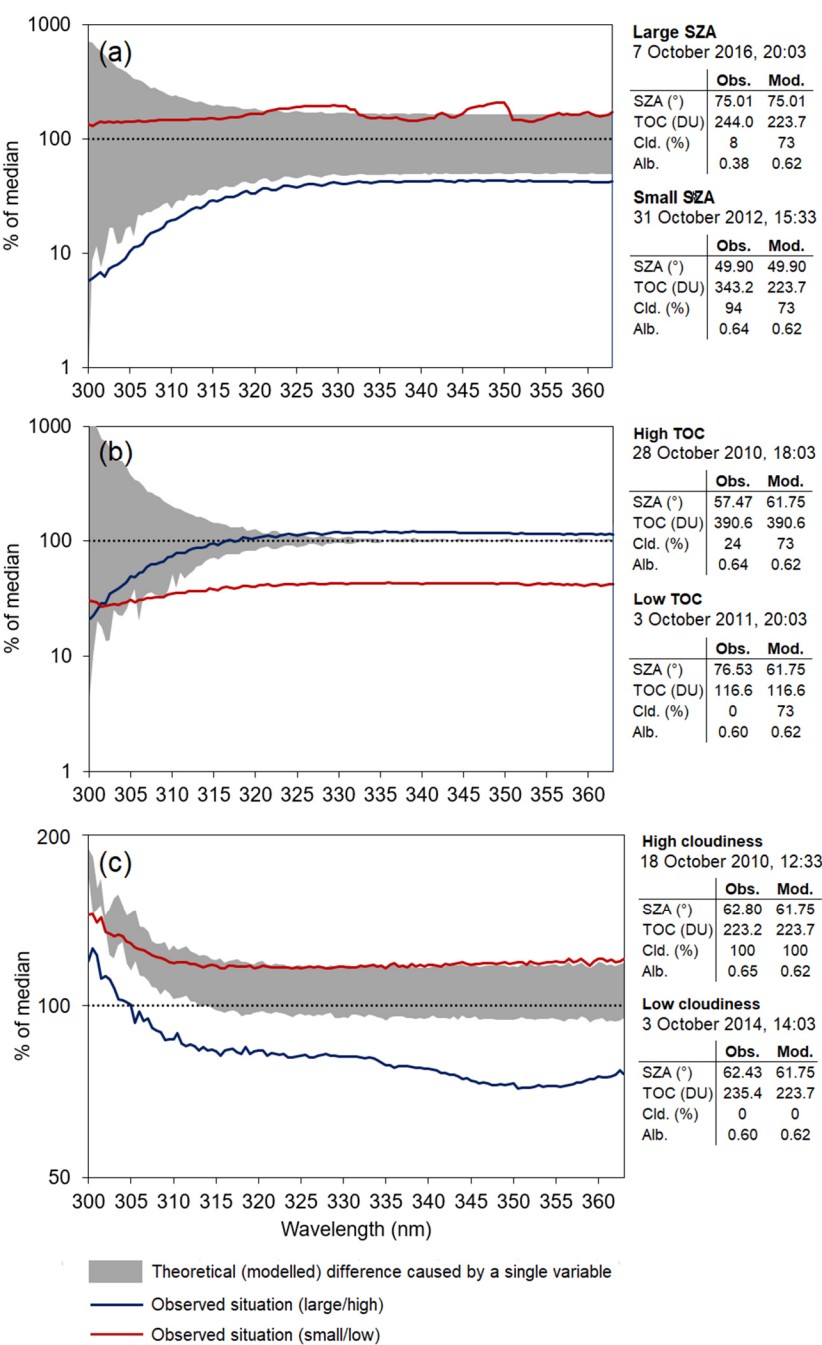

**Figure 10: Spectral UV irradiance relative to the monthly median on eight selected October days and the potential impact of selected variables (grey area) with extremely large/high or small/low (a) SZA, (b) TOC, and (c) cloud cover at Marambio Base in 2010–2020.**

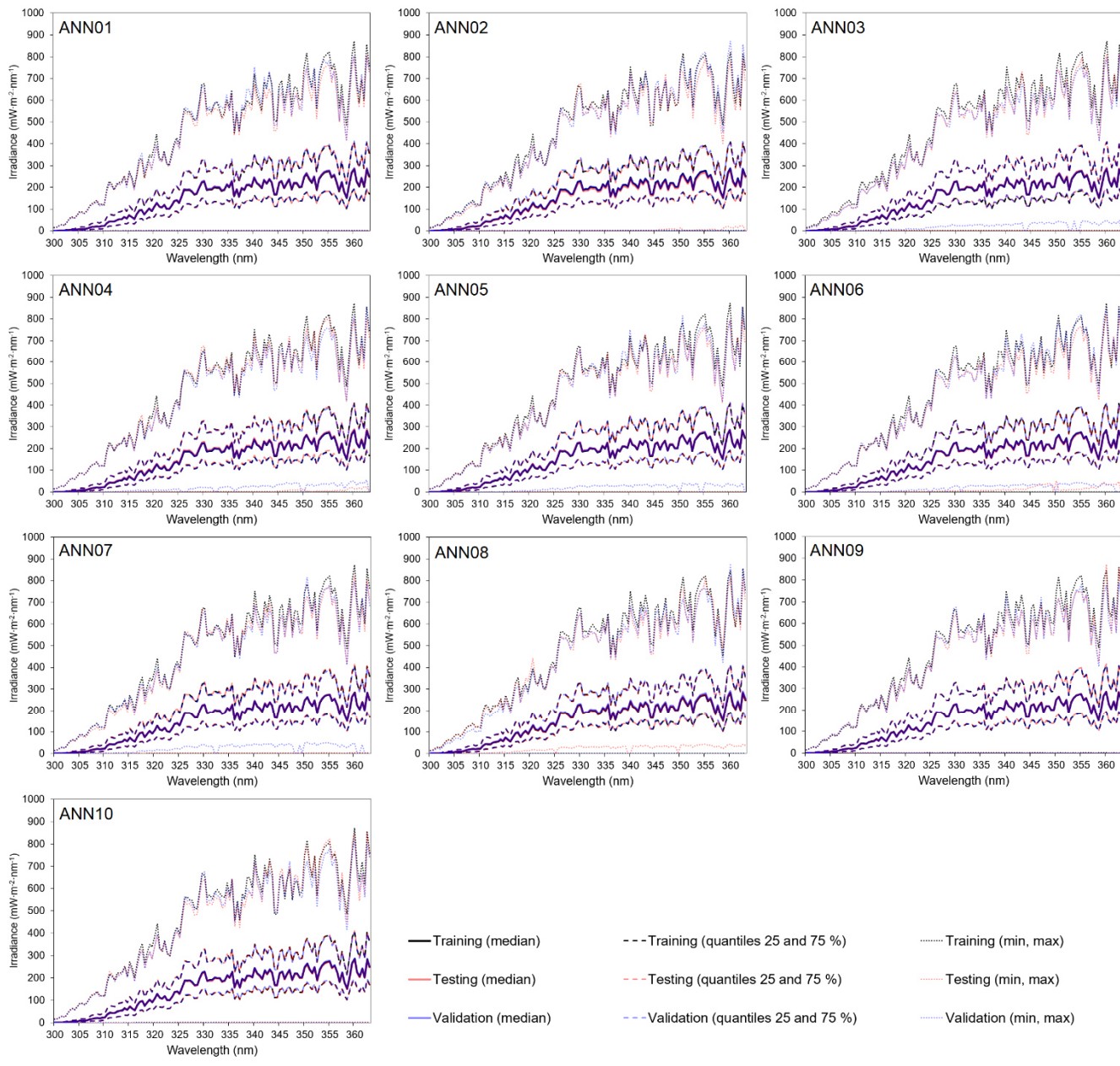


**Figure A1: Statistical characteristics of the spectral UV irradiance in training, testing and validation subsets of the individual ANN models.**

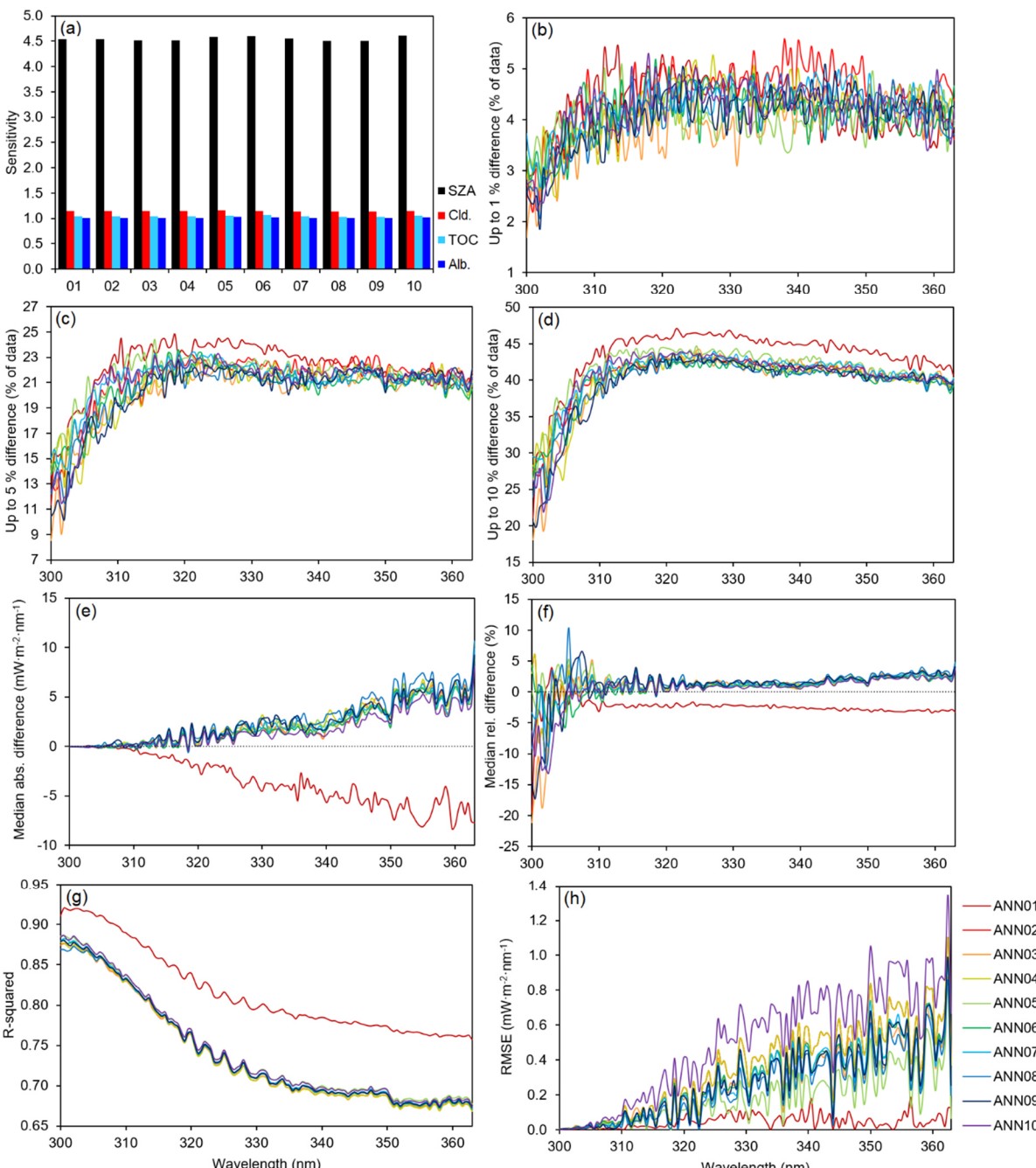

Figure A2: Development and validation of the 10 artificial neural network models, where (a) is the sensitivity of individual models to given variables; (b), (c) and (d) shows the amount of modelled data within 1, 5, and 10 % difference from observations, respectively; (e) and (f) are the absolute and relative median differences of the modelled data from the observations; (g) shows the R-squared, and (h) is the root mean square error of the individual models.