# Peer review of "Assessment of spectral UV radiation at Marambio Base, Antarctic Peninsula"

_EGUsphere, 2022_

## Author Comment (AC1)

**Author's response to comments of Referee #1**

**AC:** We thank the anonymous referee for the detailed review of our manuscript.

**RC1:** The study aims to assess the relative effect of factors affecting surface spectral radiation, including SZA, total ozone, cloudiness, and albedo. The dataset is from Brewer spectrophotometer measurements at Marambio Base from the Antarctic Peninsula. Spectral measurements at those latitudes are rare and thus the spectral UV time series 2010-2020 is very valuable, and the manuscript gives a good overview of the statistical distribution of the data. The idea of studying the effect of each UV affecting parameter is not new as such, and that has been studied in several publications previously. However, in this study, the use of neural network makes the study interesting and unique, in addition to the Antarctic location. Unfortunately there is one clear mistake in the applied inputs to the model: the surface albedo data can not be taken from the OMUVB product. In the OMUVB product, the albedo is not the actual measured albedo, but it is the albedo climatology which is used to calculate the OMI UV data. I think the paper can not be published before this has been taken into account, either by excluding the part of the study related to albedo effect or by using other data reflecting the actual albedo situation of the site. I have also a concern on the quality and homogeneity of the time series, as there was so few calibrations (see specific comments here below). In addition I don't understand what is the purpose of Figure 11 and text related to that figure (please see specific comments here below).

I am not familiar with ANN modeling, but I have the impression, that some additional information could be given related to the ten different models which were used (Line 140). How did these models differed from each others.

**AC:** Thank you for valuable comments and suggestions. The surface albedo interpretation part will be taken out of the study. The albedo climatology will only be kept neural network model, as it presents a statistically significant contribution to its accuracy. We have prepared Appendix A (at the end of the Answer sheet, we will also include it in the manuscript) explaining in detail how we built the ten ANN models and showing the differences between their performance.

Please find our answers to the individual specific comments below.

**RC1:** Specific comments:
Line 28, Reference UNEP2010 could be changed to a more recent one, eg., UNEP2019

EEAP. 2019. Environmental Effects and Interactions of Stratospheric Ozone Depletion, UV Radiation, and Climate Change. 2018 Assessment Report. Nairobi: Environmental Effects Assessment Panel, United Nations Environment Programme (UNEP) 390 pp. https://ozone.unep.org/science/assessment/eeap

**AC:** Thank you. We will change the reference to Barnes et al. (2022), which is the the latest UNEP update, where they also talk about matters covered in the introduction.

**Ref.:** Barnes, P. W., Robson, T. M., Neale, P. J., Williamson, C. E., Zepp, R. G., Madronich, S., Wilson, S. R., Andrady, A. L., Heikkilä, A. M., Bernhard, G. H., Bais, A. F., Neale, R. E., Bornman, J. F., Jansen, M. A. K., Klekociuk, A. R., Martinez-Abaigar, J., Robinson, S. A., Wang, Q.-W., Banaszak, A. T., Häder, D.-P., Hylander, S., Rose, K. C., Wängberg, S.-Å., Foereid, B., Hou, W.-C., Ossola, R., Paul, N. D., Ukpebor, J. E.,

Andersen, M. P. S., Longstreth, J., Schikowski, T., Solomon, K. R., Sulzberger, B., Bruckman, L. S., Pandey, K. K., White, C. C., Zhu, L., Zhu, M., Aucamp, P. J., Liley, J. B., McKenzie, R. L., Berwick, M., Byrne, S. N., Hollestein, L. M., Lucas, R. M., Olsen, C. M., Rhodes, L. E., Yazar, S., and Young, A. R.: Environmental effects of stratospheric ozone depletion, UV radiation, and interactions with climate change: UNEP Environmental Effects Assessment Panel, Update 2021, Photochem. Photobiol. Sci., 21, 275–301, 2022.

**RC1:** line 30: incident UV irradiance → incident short wavelength UV irradiance. Or change the wording to describe that it prevents the short wavelength UVC and UVB part of the UV spectrum.

**AC:** Agreed. We will change the wording to "incident short wavelength UV irradiance", because the UV radiation spectrum division is introduced later in the text.

**RC1:** Line 33: The recent positive stratospheric ozone trends….
Please specify in which part of the globe? Or global mean?

**AC:** Thank you. This should mean "southern polar regions". It will be changed in the revised version of the manuscript.

**RC1:** Line 88: Please specify the method for calibration against the reference instrument. Usually a Brewer is calibrated using 1kW or 200W lamps. Please specify also the traceability of the irradiance scale. Any comparison with the PMOD-World reference QASUME spectroradiometer? Either between the IOS and QASUME or between Marambio's Brewer and QASUME? Only two calibration for 11 years of measurements is very little. There is a possibility that the response of the instrument has changed unexpectedly between the calibrations. Do you have any records to check if the instrument has been stable over the years? How did you take into account the change in calibration: linear interpolation between the two calibrations or stepwise? How much was the difference between the two calibrations? Did you perform any final calibration in 2020? Please explain these points in the text.

**AC:** The instrument was serviced each year in January or February by specialists from CHMI and International Ozone Services, Inc., Canada. Each year, a calibration has been performed using three to five travel 50W lamps, which were calibrated right before the departure to Marambio Base using the B184 Brewer spectrophotometer in the CHMI Solar and Ozone Observatory in Hradec Králové, and three 1000W lamps S1450, S1451, and S1542 calibrated in the World Radiation Center in Davos. Moreover, in 2012 and 2016, the B199 spectrophotometer was calibrated against the world traveling standard B017 and spectrally calibrated with the use of HG and CD spectral lamps directly at Marambio Base. The spectrum was calibrated in 2012, 2016, and also in 2019. A final calibration was performed in 2020 both at Marambio and after the instrument returned to the Solar and Ozone Observatory in Hradec Králové. The yearly calibration results, which were taken in account stepwise, yielded a maximum difference of -7 % in 2014. The mean absolute annual difference in the 2010–2020 period was 4.1 %. The standard lamp ratios R6, R5, and results from Dead time and Run stop test are additional parameters to monitor for checking of the instrument stability. These information are saved in the instrument checklist at the Solar and Ozone Observatory in Hradec Králové.

We will add all this information to the revised version of the manuscript.

**RC1:** Line 96: Which method did you use for detection of spikes and wavelength shifts? Please add references.

**AC:** We have used the spike-detection method by Meinander et al. (2003), and the wavelength shift was analyzed based on Fraunhofer lines using the SHICrivm software package (https://www.rivm.nl/en/uv-ozone-layer-and-climate/shicrivm). The information will be added to the text of the manuscript.

**Ref.:** Meinander, O., Josefsson, W., Kaurola, J., Koskela, T., and Lakkala, K.: Spike detection and correction in Brewer spectroradiometer ultraviolet spectra, Opt. Eng., 42, 1812–1819, 2003.

**RC1:** Line 120: What did you use as inputs to LibRadtran? Please add the info in the text. And how did you calculated the CMF?

**AC:** The input parameters for libRadtran were as follows: day of year, solar zenith angle, albedo climatology from OMI, and total ozone column from B199 Brewer at Marambio.

Cloud Modification Factor (CMF) was calculated as the ratio between observed and clear-sky irradiance for each wavelength (like in, for instance, Lindfors et al., 2007), then the weighted mean using the clear-sky intensities as weights was derived for each spectrum. CMF was determined from the ground based spectral UV irradiance data from B199, and the theoretical clear-sky spectral UV irradiance was estimated using the one dimensional DISORT solver of the libRadtran radiative transfer package.

We will add the information on both libRadtran and CMF in the revised manuscript.

**Ref.:** Lindfors, A., Kaurola, J., Arola, A., Koskela, T., Lakkala, K., Josefsson, W., Olseth, J. A., and Johnsen, B.: A method for reconstruction of past UV radiation based on radiative transfer modelling: Applied to four stations in northern Europe, J. Geophys. Res., 112 (D23201), 1–15, 2007, DOI: doi:10.1029/2007JD008454.

**RC1:** This is not true. The albedo is the albedo climatology used in calculation of the OMUVB product. Please see my General comments.

**AC:** Agreed. Relevant changes (correct data product description, omittance of albedo interpretation) will be made in the entire manuscript.

**RC1:** Line 154: Do you mean the 80% of data was within +-25%?

**AC:** Yes, instead of "80 % of the data did not exceed approximately ±25 %" we will change the wording in the revised version of the manuscript as follows "80 % of the modelled data was within approximately ±25 %".

**RC1:** Line 156: What is R2?

**AC:** It is the determination coefficient (R-squared). The correct explanation will be added in the revised version of the manuscript.

**RC1:** Lines 156-157: What do you mean by shares variability and shared variability? Do you mean shared variance, covariance? Please use other wording.

**AC:** Determination coefficient represents shared variance between the two datasets (this site provides a very nice and straightforward explanation: https://www.investopedia.com/terms/c/coefficient-of-determination.asp, but it can also be found in McClave and Dietrich, 1991). Based on this statement the sentence will be rephrased in the revised version of the manuscript, and the McClave and Dietrich (1991) citation will be added.

**Ref.:** McClave, J. T. and Dietrich, F. H.: Statistics, San Francisco, CA: Dellon Publishing Company, 928 p., 1991.

**RC1:** Line 177: I suggest that you include the info of how many order of magnitude the UV irradiance changes between e.g., 300 nm and 400 nm.

**AC:** Thank you for the suggestion. Since our measurements only cover the interval between 300–363 nm, we will chose 305 and 340 nm as examples here, as we cover these wavelengths also further in the text (they are contrasting when it comes to absorption by ozone, and they are also frequently used in literature, which allows a comparison of results). The information that between these two wavelengths, the median UV irradiance increased approximately 25 times in summer and over 100 times in early spring and late fall, will be added in the revised version of the manuscript.

**RC1:** Line 179: ……..it is steeper in low-SZA months…….. Explain why (longer atmospheric path – more ozone absorption)

**AC:** The statement that "it happens because at low SZA, the atmospheric path is shorter, so less UV radiation is absorbed by ozone and other atmospheric gases" will be added to the revised manuscript.

**RC1:** Line 185: What do you mean with "overall median". Yearly median? Something else?

**AC:** It is the median value from all available measurements over the entire study period (calculated separately for each wavelength). We will clarify it in the revised version of the manuscript.

**RC1:** Line 192: The vertical profile of atmospheric ozone affects the absorption in the optical path. This info could be added somewhere, and possibly discussed.

**AC:** Thank you for the suggestion and the interesting idea, which we may focus on in further research and do a couple of analyses! In the revised version of the manuscript, we will only add a short note to Section 3.2 (explanatory variable effects), to the end of the paragraph about ozone as follows: "However, the shape of vertical ozone profile may play a substantial role in UV radiation absorption in the optical path, as different vertical distributions of ozone may lead to similar TOC values".

**RC1:** Lines 193-197: I think you should, in a couple of sentences, describe the effect of the Brewer-Dobson circulation and the lack of sunlight in winter → ozone accumulates in polar region, but there is no sunlight for the photochemical ozone destruction which then leads to high TOC values at the end of the winter. You should also explain the year to year variation in Antarctic ozone depletion, otherwise it is quite strange that you find within during the same week the highest and the lowest ever measured TOC (6 November 2011, 3 November 2015).

**AC:** Agreed, thank you for the suggestion. We will add a short explanation, together with relevant citation, to the revised version of the manuscript:

"The yearly cycle, as well as the short and long term fluctuations of TOC in the coastal Antarctic region are conditioned by both chemical and dynamical influences, while the chemical ones (like the catalytic reactions with the contribution of man-made chemicals) are now quite well understood (e.g., Solomon, 1999). The dynamical influences include the Brewer-Dobson circulation, which causes the poleward transport of ozone from the tropic to the poles, and subsequent accumulation of ozone in the polar regions in winter, as no UV radiation is present to induce ozone loss (Weber et al., 2011). Ozone depletion through catalytic reactions starts in early spring and low TOC values are present till the breakdown of the polar vortex, which is caused by the dynamical effect of planetary waves and has much year-to-year variability (e.g., Shepherd, 2008), so it was possible to observe both the absolute ozone minimum and maximum in one month (November), only in different years."

**Ref.:**
- Shepherd, T. G.: Dynamics, Stratospheric Ozone, and Climate Change, Atmos. Ocean, 46, 117–138, 2008.
- Solomon, S.: Stratospheric ozone depletion: a review of concepts and history, Rev. Geophys., 37, 275–316, 1999.
- Weber, M., Dikty, S., Burrows, J. P., Garny, H., Dameris, M., Kubin, A., Abalichin, J., and Langematz, U.: The Brewer-Dobson circulation and total ozone from seasonal to decadal time scales, Atmos. Chem. Phys., 11, 11221–11235, 2011.

**RC1:** Line 199: Where is the site of Lachlan-Cope (2010) located?

**AC:** Lachlan-Cope (2010) used latitudinal means, because his work was based on satellite data. Corresponding changes (clarification) will be added in the revised version of the manuscript.

**RC1:** Lines 200-2004: I am not convinced why do you compare with these sites, and about the reasons for the differences. Is there differences in land-sea surroundings, glaciers ...topography, other meteorological reasons, typical synoptic scale phenomena?

**AC:** All the three mentioned studies used satellite data, which covered, in the case of Lachlan-Cope (2010), the latitude of Marambio, and in the case of the other two mentioned studies, a pixel directly containing the location of Marambio, therefore no big differences in e.g. topography shall be present.

The results in our manuscript, which gave a different annual cycle than the three above-mentioned studies, are not statistically significant (mentioned in the text), and therefore they are non-conclusive, affected likely only by the high cloud cover variability. We will clarify the use of satellite data in the other studies and further stress the non-conclusiveness of our results, which were not statistically significant.

**RC1:** Lines 228-229: Alongside atmospheric gases, the surface UV irradiance is also affected by the absorption in the troposphere, which is in this study represented by the cloud cover.
How about tropospheric O3, SO2? What do you mean by "which is in this study represented by the cloud cover"?

**AC:** This was only meant to be an introduction sentence. In the revised manuscript, we will change the wording to the simple "Another parameter, important for the UV radiation attenuation, is cloud cover." Unfortunately, it was not within the scope of the study to cover also tropospheric ozone or sulfur dioxide, although it could be an interesting idea for future research. Particular information about this possible further research will be added it to the conclusion paragraph of the revised manuscript.

**RC1:** Lines 245->end of the section. Please specify what do you mean by "very high UV irradiances". Very high compared to what? Or higher than a certain limit value?

**AC:** Very high UV irradiances at 305 and 340 nm were defined as the highest 10 % of all recorded values. The statement will be changed in the revised version of the manuscript in order to be more understandable.

**RC1:** Line 277: ozonosphere This term doesn't exist.

**AC:** Thank you for the notification. The term will be removed in the revised version of the manuscript, replaced by "ozone layer".

**RC1:** Line 285: in low-SZA conditions →Should this be in high-SZA..?

**AC:** We don't think so, as the radiation intensity at very short UV wavelengths decreases faster at low SZA, as shown in the figure taken from Kerr and Fioletov (2008), which is attached (x-axis is the cosine of SZA). However, to avoid possible confusion, we will rephrased the statement in the revised version of the manuscript.

[Figure]

Fig. 2    Measurements of spectral irradiance at two wavelengths in the UV spectrum. Measurements were taken at Toronto between January 2002 and June 2003. There is strong absorption by ozone at 300 nm (a) and very little absorption at 324 nm (b). The clustering near the top of the 324 nm data set indicates clear-sky conditions. There is no similar clustering at 300 nm since the effects of ozone variability mask cloud variability. The fall-off of irradiance with decreasing cos(sza) is significantly sharper at 300 nm than that at 324 nm, where it is nearly linear. Note that the irradiance values are about two orders of magnitude larger at 324 nm than at 300 nm.

**Ref.:** Kerr, J. B. and Fioletov, V. E.: Surface Ultraviolet Radiation, Atmos. Ocean, 46, 159–184, 2008.

**RC1:** Lines 309-317 I think this discussion should be excluded if actual albedo values are not used (See general Comments)

**AC:** Agreed. This part will be taken out from the revised version of the manuscript, alongside any other conclusions drawn from albedo effects (in summary, abstract, etc.).

**RC1:** From line 318 until the end of the section. I don't understand what has been done. Why didn't you keep in the model the same values than in the observation (Figure 11)?

**AC:** This part of the study has two purposes: first to show the differences between the high and low values of a given parameter (through the model values, where all other variables are fixed to their monthly medians). The second purpose was to show the dissimilarity between the model outputs and observations, which are meant to express the importance of one (model) vs. all (observation) studied parameters. If we ran the model with the observed values, it would have been a validation, which was already performed before (fig. 3). This way, we could compare the dissimilarities between the model and observations, which were caused by the differences in input variables. Based on the above-mentioned statement, we will expand the explanation in the Results section and clarify the purposes and the interpretation of the figure in the revised version of the manuscript. Moreover, the wording will be changed also in the Method section, to make it clear this was not a model validation but actually a modelling experiment, a study of the dissimilarities between the modelled and observed values.

**RC1:** Figures 7-9 caption: Include the information if the effect is the median effect, and if yes, to which quantity do you make the change (actual value during the measured spectrum?).

**AC:** Thank you. Please note that we have presented the mean change; the relative values were calculated with regards to the monthly median spectra. This information will be added to the captions of related figures in the revised version of the manuscript.

**RC1:** Figure 10. I think you should exclude this plot, as the albedo is not the actual one. You can not make any conclusion of it's influence.

**AC:** Thank you. This plot will be excluded, alongside the albedo panels of Fig. 4, 5 and 11 (now Fig. 10) in the revised version of the manuscript.

**Appendix A**

**Artificial Neural Network model development and validation**

Out of the ten ANN models we built, nine (ANN02 to ANN10) behaved in a similar way, while one (ANN01) was different. The differences between the models did not result from the ANN setting, which remained the same, but occurred due to the random initialization of the models and the random split of the dataset to training (70 %), testing (15 %), and validation (15 %) subsets. As seen from Fig. A1, the model ANN01 had the most data within ± 5, respective ± 10 % from observations, and it had the largest R-squared and lowest RMSE out of all ten models. However, the model was biased toward underestimation of UV irradiance throughout most of the spectrum.

For the purpose of the study, it was best to choose a model with the best precision, i.e. the lowest variability of results, highest R-squared and lowest RMSE (model ANN01). Also, it was possible to tackle the bias present within the model using a simple median correction described in the manuscript in section 2.4.

[Figure]

**Figure A1.** Development and validation of the 10 artificial neural network models, while (a) is the sensitivity of individual models to given variables; (b), (c) and (d) shows the amount of modelled data within 1, 5, and 10 % difference from observations, respectively; (e) and (f) are the absolute and relative median differences of the modelled data from the observations; (g) shows the R-squared, and (h) is the root mean square error of the individual models.

---

## Author Comment (AC2)

**Author's response to comments of Referee #2**

**AC:** We thank the anonymous referee for the review of our manuscript.

**RC2:** I read an interesting article which aims to investigate the effect of solar zenith angle, ozone, cloud cover and surface albedo on spectral UV radiation at Marambio Base, Antarctic Peninsula. UV irradiance measurements come from a double Brewer spectrophotometer for the period 2010-2020. The effects of the different parameters on surface UV irradiance are studied using a neural network model that has been developed for this purpose. My recommendation for this article is to accept for publication after clarifying better what Figure 11 aims to show, as I had also recommended during my quick review, and after clarifying the neural networking explained in section 2.4.

**AC:** Thank you for your comments and suggestions. Concerning artificial neural networks (ANN), we would like to mention that dozens of cross-validated tests of randomly initialized ANN with random division of the whole dataset to training (70% of data), validation (15%) and testing (15%) subsets were carried out. The aim of these tests was to set the appropriate complexity of ANN, corresponding to the complexity of underlying relation between chosen predictors (SZA, TOC, cloudiness, albedo climatology) and predictands (spectral UV intensities). These tests revealed that about 22 neurons in the hidden layer are (quasi)optimal to avoid both over- and under-parametrisation of the relationship. The advantage of this approach is that it derives the appropriate complexity of ANN directly from real (measured) data and is not limited by an a priori assumption about the shape of the regression function. In addition, ANN are capable of modelling relationships that are difficult to describe analytically and from this point of view, they are more general than the analytical approach. Then, an ensemble of 10 neural networks, each with 22 neurons in the hidden layer were trained, again with random division of the dataset to training (70%), validation (15%) and testing (15%) subsets and with random initialization of the networks. The overtraining of the networks was tackled by adding a stopping condition: the error improvement lower than 0.0000001 in the window of 200 cycles, while the error function was defined as the sum of squares.

Based on the above-mentioned information, we will expand the explanation and interpretation of particular plots/figures in the revised version of the manuscript. Moreover, we are providing the new plots stating how we built the models and showing the differences between them (see Appendix A, which we will also upload in the manuscript).

**RC2:** Specific comments:
Lines 139-161: I find difficult to understand how the ten models were built and how the best neural network model was selected each time. Given that most studies use multiple regression modelling to quantify the contribution of each atmospheric parameter, lines 140-142 trigger the question how much different would the calculations from a multiple regression model be? A supplement with explanations on the neural model procedures, and comparison with estimations from a multiple regression model would help.

**AC:** Please note that ANN derive the relationship between predictors and predictands directly from the input data, so they are not limited by any a priori assumptions about the shape of the dependence. An example of the potential assumption based on Lambert-Bourger-Beer Law can be found e.g., in Antón et al. (2005). Moreover, with sufficient complexity of the network (the number of hidden layers and the number of neurons in them), they are able to simulate practically any dependency, even that one that is difficult to describe analytically. In this sense, they are more general than classical regression approach. A direct comparison of the classic regression approach, based on Beer's law, and the neural network is in this particular case not possible, as an independent variable (the intensity of

UV radiation at the top of the atmosphere) enters the regression relationship described in Antón et al. (2005), but it is not an input parameter of our ANN. Thus, both models are based on different sets of input data. Moreover, the neural model simulates the UV spectrum as a whole, while the regression approach, using Beer's law, calculates the intensities independently and for each wavelength separately. The neural network can therefore better simulate the interdependencies between UV radiation intensities at different wavelengths, which is especially important in a situation where the data contain inaccuracies or noisy components. This is where the two methods differ considerably, and a specific study would have to be designed to allow a proper comparison.

We agree, however, that the information we provided on the ANNs in the manuscript was limited. Therefore, based on the above-mentioned information and Appendix A, we will stress ANN modelling more in the revised version of the manuscript.

Ref.: Antón, M., Cancillo, M. L., Serrano, A., and García, J. A..: A Multiple Regression Analysis Between UV Radiation Measurements at Badajoz and Ozone, Reflectivity, and Aerosols Estimated by TOMS, Phys. Scripta, 118, 21–23, 2005.

**RC2:** Low albedo case (fig. 11d): the Obs. Alb. value is indeed low (0.37), but the Mod. Alb. value is 0.81, which is not low. Please check.

**AC:** Thank you for the comment. This was indeed an error, it should have been 0.37. However, based on the comments from Referee 1, the albedo panel as well as related calculation will be removed from the manuscript.

**Appendix A**

**Artificial Neural Network model development and validation**

Out of the ten ANN models we built, nine (ANN02 to ANN10) behaved in a similar way, while one (ANN01) was different. The differences between the models did not result from the ANN setting, which remained the same, but occurred due to the random initialization of the models and the random split of the dataset to training (70 %), testing (15 %), and validation (15 %) subsets. As seen from Fig. A1, the model ANN01 had the most data within ± 5, respective ± 10 % from observations, and it had the largest R-squared and lowest RMSE out of all ten models. However, the model was biased toward underestimation of UV irradiance throughout most of the spectrum.

For the purpose of the study, it was best to choose a model with the best precision, i.e. the lowest variability of results, highest R-squared and lowest RMSE (model ANN01). Also, it was possible to tackle the bias present within the model using a simple median correction described in the manuscript in section 2.4.

[Figure]

**Figure A1.** Development and validation of the 10 artificial neural network models, while (a) is the sensitivity of individual models to given variables; (b), (c) and (d) shows the amount of modelled data within 1, 5, and 10 % difference from observations, respectively; (e) and (f) are the absolute and relative median differences of the modelled data from the observations; (g) shows the R-squared, and (h) is the root mean square error of the individual models.

---

## Author Response (AR2)

**Author response to referee comments**

1. Referee comments (RC3) are in **BLACK**
2. Authors' answers (AC) are in **BLUE**
3. Changes in the manuscript (MC) are in **PURPLE**; the lines refer not to the original, but to the revised version of the manuscript.

**Author's response to comments of Referee #3**

**AC:** We thank the anonymous referee for the review and all the comments and remarks you made. We addressed all major and minor comments below, and stated what changes were made to the manuscript based on them.

**RC3:** This is a review of a revised version of a manuscript titled "Assessment of spectral UV radiation at Marambio Base, Antarctic Peninsula" by Čížková et al. I have also reviewed the response of the authors to the comments of two reviewers pertaining to the original version of the manuscript. I confirm that that the authors have addressed the reviewers' concerns appropriately. Unfortunately, I have found additional major flaws in the revised manuscript, which must be addressed before the manuscript can be accepted for publication. Most importantly, data of the Brewer spectrophotometer and the complementing model calculations should have been reported as "spectral irradiance", not "irradiance". Furthermore, data presented in the manuscript are too low by at least two orders of magnitude. Most data therefore have to be reprocessed. In addition, data presented in Figures 7–9 show large artefacts related to the artificial neural network model. Most notably, Figure 8c indicates that UV irradiance depends on total column ozone at wavelengths larger than 350 nm when in fact (and confirmed by my own calculations) the ozone absorption cross section in this wavelength range is too small to have a noticeable effect on UV irradiance. The dataset is worth publishing, considering that there are only a few sites in Antarctica that provide spectral UV data. However, my "major" and "minor" comments should be addressed first, and the language should be improved also. I have included several pages related to language with suggestions to improve the text and make it more readable.

**\*\*\*Major comments**

**RC3:** \*\*Data are not provided in the correct quantity and the magnitude of the results is incorrect

All results are provided as "irradiance" in units of mW m-2. A Brewer MK III measures spectral irradiance, not irradiance, in units of mW m-2 nm-1 (with a spectral resolution (bandwidth) of 0.5 nm according to the authors; although, to my knowledge, the resolution of this instrument is 0.6 nm). If the authors provide results in irradiance instead of spectral irradiance, the wavelength interval over which their spectral data were integrated to get irradiance has to be specified. Furthermore, irradiances presented by the authors seem to be off by several orders of magnitude. For example, for a solar zenith angle (SZA) of 60°, I would expect a spectral irradiance of 335 mW m-2 nm-1 at 363 nm for a spectroradiometer with a 1 nm wide triangular slit function. The "irradiance" data presented by the authors in Figure 4 max out at about 0.8 mW m-2. This is different from the correct result (presuming that irradiance data were integrated over a wavelength interval of 1 nm) by a factor of more than 400. Even if data had only been integrated over 0.5 nm (the resolution and sample-step of the Brewer) instead of 1 nm, as in my calculations, the results would be off by two orders of magnitude. As a result of these errors, all subsequent results presented in absolute terms (irradiance in mW m-2) and model results (which are tied

to measurements) in Figure 7–10 are incorrect. All absolute values in the manuscript have to be recalculated to correct for these errors.

**AC:** Thank you for noting the discrepancies. The Brewer spectrophotometer measures in the sampling interval of 0.5 nm, and the measured wavelength interval (full width half maximum) is 0.6 nm. We will clarify this in the text. The reason of the errors in UV irradiances was that in the original paper, the units were in $W \cdot m^{-2} \cdot nm^{-1}$, not in $mW \cdot m^{-2} \cdot nm^{-1}$.

**MC:** The units were changed to $mW \cdot m^{-2} \cdot nm^{-1}$ throughout the entire manuscript, of course including all the figures. Also, the wording was changed from "irradiance" to "spectral irradiance" where applicable throughout the manuscript. The clarification of Brewer spectrophotometer sampling interval and FWHM was added in lines 101–102.

**RC3:** **The artificial neural network (ANN) is not described.

It is stated in line 156 that the ANN model is a "perceptron" model, but no other details are provided. Please provide information on this model, either a reference or a description on how it works. Without any detail, it is just a black box. Journal papers like this require a description that would allow the reader to reproduce these model calculations.

**AC:** Thank you for the comment. The model was built as follows, using the TIBCO Statistica software (TIBCO, 2023):

First of all, many crossvalidated tests of randomly initialized neural networks with a random division of the whole dataset to training (70 % of data), validation (15 %) and testing (15 %) subsets were carried out (the process is also described in for example Malik et al., 2022). The aim of these tests was to find the best set of predictors and to establish the appropriate complexity of the neural network, corresponding to the complexity of underlying relations between predictors (SZA, TOC, cloud cover, albedo) and predictands (spectral UV irradiance). These tests revealed that about 22 neurons in the hidden layer are (quasi)optimal to avoid both over- and underparametrisation of the relation.

Then, an ensemble of 10 neural networks with logistic activation functions, each with 22 neurons in the hidden layer were trained, again with random division of the dataset to training (70%), validation (15%) and testing (15%) subsets and with random initialization of the networks. The "early stopping" measure against overtraining is already described in the text.

- TIBCO Statistica® User's Guide: Statistica Automated Neural Networks (SANN) - Neural Networks Overview, https://docs.tibco.com/pub/stat/14.0.0/doc/html/UsersGuide/GUID-F60C241F-CD88-4714-A8C8-1F28473C52EE.html, last access: 3 February 2023.

The advantage of the ANN approach is that it derives the appropriate complexity of the neural network directly from even a relatively simple real data, and it is not limited by an a priori assumption about the shape of the regression function. In addition, neural networks are capable of modeling relationships that are difficult to describe analytically and from this point of view, they are more general than the analytical approach (e.g., Barbero et al., 2006; Malik et al., 2022).

Therefore, ANN models are commonly used in atmospheric sciences including UV radiation climatology, for example for purposes like forecasting or reconstructions, and they perform reasonably well. Here you

can find the citations of several works that successfully illustrate the use of ANN modeling in UV climatology:

- Barbero, F. J., López, G., and Batlles, F. J.: Determination of daily solar ultraviolet radiation using statistical models and artificial neural networks, Ann. Geophys., 24, 2105–2114, 2006.
- Feister, U., Junk, J., Woldt, M., Bais, A., Helbig, A., Janouch, M., Josefsson, W., Kazantzidis, A., Lindfors, A., den Outer, P. N., and Slaper, H.: Long-term solar UV radiation reconstructed by ANN modelling with emphasis on spatial characteristics of input data, Atmos. Chem. Phys., 8, 3107–3118, 2008.
- Latosińska, J. N., Latosińska, M., and Bielak, J.: Towards analysis and predicting maps of ultraviolet index from experimental astronomical parameters (solar elevation, total ozone level, aerosol index, reflectivity). Artificial neural networks global scale approach, Aerosp. Sci. Technol., 43, 301–313, 2015.
- Raksasat, R., Sri-iesaranusorn, P., Pemcharoen, J., Laiwarin, P., Buntoung, S., Janjai, S., Boontaveeyuwat, E., Asawanonda, P., Sriswasdi, S., and Chuangsuwanich, E.: Accurate surface ultraviolet radiation forecasting for clinical applications with deep neural network, Sci. Rep.-UK, 11 (5031), 1–12, DOI: 1038/s41598-021-84396-2, 2021.

ANNs are also used in various studies concerning solar radiation in general, as shown for example in this review, which even claims ANN can provide better results than conventional methods:

- Yadav, A. K. and Chandel, S. S.: Solar radiation prediction using Artificial Neural Network techniques: A review, Renew. Sust. Energ. Rev., 33, 772–781, 2014.

A more recent review of the use of ANNs in solar radiation climatology shows ANNs are still a widely used method, with the statements confirming that when extrapolation is avoided (which was done in our case), ANNs provide results of similar, or even better quality to conventional methods:

- Malik, P., Geholt, A., Singh, R., Gupta, L. R., and Thakur, A. K.: A Review on ANN Based Model for Solar Radiation and Wind Speed Prediction with Real-Time Data, Arch. Comput. Method. E., 29, 3183–3201, 2022.

**MC:** The ANN model description was extended (lines 169 – 193), and the reference to the User's Guide of the program we used was added (line 188 and lines 685–686). We also cited the relevant works we mentioned above (lines 176–177 in the text).

**RC3:** **The spectral UV measurements are insufficiently described.

For example, how many spectra were measured per day? Are these spectra distributed equally throughout the day (otherwise the average and median of daily measurements would be skewed). What is the uncertainty of the measurements? Have the measurements been independently validated, for example, by participating in an intercomparision campaign with a reference spectroradiometer.

**AC:** Thank you for the comment.

There were between one and 29 spectral measurements taken during each day with observations, the distribution is shown in Fig. 1 below.

[Figure]

Fig. 1. Number of days with a certain number of spectral UV measurements per day at Marambio Base, 2010–2020.

The temporal distribution of the measurements was more or less even in small-SZA months and centered around noon in large-SZA months, as seen from Fig. 2.

[Figure]

Fig. 2. Temporal distribution of spectral UV measurements in individual months at Marambio Base, 2010–2020.

Therefore, the measurements are centered around low SZAs in summer months, and around mean SZAs in spring and fall. The median SZAs are shifted toward higher than modal values especially in the lowest-SZA months, as seen in Fig. 3, and due to this non-normal distribution, median was considered a more representative measure of central tendency than mean.

[Figure]

| | AUG | SEP | OCT | NOV | DEC | JAN | FEB | MAR | APR |
|---|---|---|---|---|---|---|---|---|---|
| Median SZA | 76.11 | 70.13 | 61.75 | 55.60 | 52.12 | 53.50 | 59.27 | 67.12 | 74.59 |

Fig. 3. The distribution of spectral UV measurements in individual months based on SZA and the corresponding median SZA of each month at Marambio Base, 2010–2020.

In the manuscript, we will include Fig. 1, 2 and 3 from this response as panels b, c and d in Fig. 2, and we will add a description similar to the one provided here.

Brewer #199 was calibrated with Canadian traveling standard Brewer #017 in Marambio in 2012 and 2016. The uncertainties of Brewer double monochromator UV spectra measurements are up to 5 %, as shown by, for example, Lakkala et al. (2008), which is comparable to our instrument as the Czech and Finnish meteorological institutes use similar calibration techniques and standards.

Ref.: Lakkala, K., Arola, A., Heikkilä, A., Kaurola, J., Koskela, T., Kyrö, E., Lindfors, A., Meinander, O., Tanksanen, A., Gröbner, J., and Hülsen, G.: Quality assurance of the Brewer spectral UV measurements in Finland, Atmos. Chem. Phys., 8, 3369–3383, 2008.

MC: The Figures 1, 2 and 3 from this response were included as panels b, c and d in Fig. 2 (line 718), and the description of the spectra was extended (lines 110–118). We also added the information about the uncertainties (line 95).

**RC3:** \*\*Figures 7-9 show artifacts of the ANN model, so some features presented in these figures are not real, and some artefacts lead to wrong conclusions.

Results should be double-checked with a radiative transfer model (which the authors apparently have because they calculate CMFs). It would be simple to model the UV irradiance as a function of SZA for each month for a fixed TOC and CMF. It would be much more convincing to have results based on a physically correct model than the author's ANN model. The following features are likely artefacts of the ANN model:

In Figure 7a, UV irradiances do not asymptotically approach zero for large SZAs but seem to have an offset. In Figure 7b, there is a spurious maximum at about SZA=45 degree. In Figure 7d, large, wavelength-dependent fluctuations are apparent at small wavelengths in particular for March and April, etc.

The greatest artefacts are apparent in Figure 8c. Because of the steep decline of the ozone absorption cross section in the Huggins band, the effect of TOC on UV irradiance becomes very small for wavelengths larger than 340 nm, and in particular for wavelengths larger than 350 nm. Figure 8c contradicts the diminishing effect of changes in TOC on UV irradiance with increasing wavelength. Here, a 10 DU change in TOC has the largest effect on UV irradiance at 363 nm (in January). Furthermore, Figure 8d shows unrealistic spikes in the relative change at about 303 and 305 nm for March and April. The relative change in Figure 8d seems to be constant (albeit small) for wavelengths larger than ~337 nm. Instead the relative change should continuously decrease between 337 and 363 nm.

To check my assertions, I modeled UV spectra for SZA=60 degree and TOC of either 300 or 310 DU (i.e., an increase by 10 DU) using the ozone absorption cross-section by Molina, which is provided up to 350 nm. As expected, the relative change decreased with increasing wavelength and was below 0.02% for wavelengths larger than 345 nm. The absolute change peaked at about 317 nm at a irradiance of 2.1 mW m-2 nm-1, decreased steadily with increasing wavelengths from there onward, and was below 0.02 mW m-2 nm-1 above 348 nm. The difference that I calculate with my radiative transfer model are greatly different both in magnitude and shape of the difference shown in Panel (c), confirming my conclusion that the results of the ANN model are incorrect.

In summary, Figure 8c gives a false impression of the effect of changes in TOC on UV irradiance.

Figure 9c shows unrealistic fluctuations below 310 nm.

In general, I find the "absolute" panels (c) in Figures 7–9 not very helpful because the structure is dominated by the Fraunhofer lines. The relative changes shown in panels (d) are really the interesting ones and the authors should focus their discussion on these panels.

**AC:** Thank you for this comment and suggestion. Unfortunately, it is impossible to use the radiative transfer model to assess the ANN performance because it requires a precise cloud cover structure data (amounts of water and ice in individual layers), which are not available to us as we only have the percentages from ERA5. Therefore, without increasing the error by attempting to deduce the nature of the clouds, we could not do more than to simulate the clear-sky irradiances.

The ANN artifacts are of course real, they belong to the downside of ANN modeling. However, despite the existence of artifacts, ANNs are in our case the best option, because they can operate with relatively simple datasets such as those available to us. Moreover, ANNs do not need any assumption about the form of (functional) dependence between predictors and predictands, as they are able to learn this

dependence directly from the data in the training process, so they can be seen as more general and flexible than conventional modelling methods. Another advantage is that ANNs are not prone to numerical problems in cases of interdependent predictions and they can work well even with data affected by noise, which is typical for empirical series derived from measured data. Also, as stated above, in climatology of solar radiation, ANN models are a quite commonly used tool. For all the above reasons, we consider ANN modeling to be an adequate method for solving the given task.

**MC:** We pointed out that model artifacts are the downside of ANN modeling (lines 177–178), and we described the artifacts in the Results section (lines 353–357 and 374–375). Due to possible confusion related to the artifacts and the Fraunhofer line domination, we also omitted the absolute change panels (c) from Fig. 7–9 (lines 741–753).

**RC3:** **Important literature is not cited and results are ignored

The assessment in line 61 that "a complex evaluation of long time series of solar UV irradiance spectra from Antarctica is still missing." is incorrect. For example, measurement of spectral UV irradiance at the South Pole, McMurdo / Arrival Heights, and Palmer Station, have been continuously performed since the early 1990s as part of the National Science Foundation's and NOAA Antarctica UV Monitoring Network, e.g., https://gml.noaa.gov/grad/antuv/. Numerous publications are based on these data and these publications also include quantitative assessments of the effects of ozone, albedo, cloud cover, and other factors on the measured UV spectra. The authors should consult the list of references in Bernhard and Stierle, 2020 (which they have cited), in particular #21, 23, 24, 28, 30, and 43, as well as https://doi.org/10.1029/2004JD004937, which includes a very detailed analysis on the wavelength dependence of cloud attenuation, and the variability in UV radiation caused by variations in total ozone at the South Pole.

**AC:** Thank you for the note, we apologize that we have omitted several important sources.

**MC:** All recommended literature was included in the manuscript. The paragraph about spectral UV radiation monitoring in Antarctica was extended (lines 55–57) and relevant results were discussed (lines 295–296, 378, and 388–391).

***Minor comments

**AC:** Thank you for all these comments, suggestions and explanations. Where no further "AC" comment is present, we accepted the suggestion and proceeded to manuscript changes.

**RC3:** The introduction contains much general information that could be cited and does not have to be included. So the introduction could be shortened substantially

**MC:** Based on this suggestion, we took out some parts of the introduction, especially the description of UV radiation effects, where we only quoted the relevant works (lines 23–27).

**RC3:** L21: Include reference to support "since its discovery in 1801."

**MC**: The following reference was added (lines 23 and 577–578): Hockberger, P. E.: A History of Ultraviolet Photobiology for Humans, Animals and Microorganisms, Photochem. Photobiol., 76, 561–579, 2002.

**RC3:** L34: Jovanović et al., 2019 is a rather obscure publication. Please add a more standard one.

**MC:** This publication was be removed from the manuscript and replaced by the following one: Velders, G. J. M., Andersen, S. O., Daniel, J. S., Fahey, D. W., and McFarland, M.: The importance of the Montreal Protocol in protecting climate, PNAS, 104, 4814–4819, 2007 (lines 32 and 685 – 686).

**RC3:** L37: the paper by Bernhard and Stierle does not report on positive trends in UV irradiance for the month of September. (Positive trends were reported for other months, but not for September).

**AC:** The sentence speaks about the recent positive trends in TOC, which lead to a decrease (negative trend) in UV radiation in September. Such trend was reported by Bernhard and Stierle, although it was not statistically significant.

**MC:** The information about the statistical insignificance was added to the manuscript (line 34).

**RC3:** 41–42: The wavelength ranges for UV-C, UV-B and UV-A radiation should be:

- UV-C: 100–280
- UV-B: 280–315
- UV-A: 315–400

**AC:** Thank you for the comment, you are right, this is the WHO standard. However, different publications give different scales (e.g., Diffey, 1990), so we will keep a note on the semi-arbitrary nature of the scale.

**MC:** The scale was changed and the WHO standard was cited (lines 38–40, 692–693).

**RC3:** L63: I am not convinced that there is a gap. The way SZA and ozone affects UV radiation are well understood. The effect from clouds is more difficult, but assessing the complex effects of clouds with cloud cover data from satellites covering a larger area will do little to capture these intricate effects. Please see also my last "major" comment.

**MC:** Based on this and the last "major" comment, we rephrased lines 60–61 to "This study aims to contribute to broaden the knowledge of spectral UV irradiance in Antarctica by…", so that the word "gap" is omitted.

**RC3:** L67: Regarding: "between the southern polar vortex and UV irradiance reaching the Antarctic continent": The paper describes measurements from one particular site, which is on an island and not the Antarctic continent. So these measurements can hardly be representative for the vast Antarctic continent.

**MC:** In order to avoid confusion, the sentence was rephrased to "…reaching the site" (line 65).

**RC3:** L93: It was already stated in the previous sentence that the instrument was calibrated in 2012 and 2016. So what does "the spectrum was calibrated in 2012, 2016" add here?

**AC:** This means that in 2012 and 2016, dispersion test with Hg and Cd spectral lamps was done, and UV response file was updated.

**MC:** We specified this in lines 88–91.

**RC3:** L96: What does R6 and R5 mean? What was ratioed? What are Dead time and Run stop tests? Why is "Dead" and "Run" capitalized?

**AC:** R6 and R5 are called "double ratios" of UV intensities and these are related with total ozone and $SO_2$ amount. The same ratios of intensities calculated from halogen lamp shows long term stability of spectrometer and differences of extraterrestrial constant. The dead time test of photomultiplier run-stop test of the slit mask are usual daily tests to check correct function of the Brewer spectrophotometer.

**MC:** The information was included in lines 95–99.

**RC3:** L100: What wavelength range does "very short wavelengths" refer to? (The range is explained further down. This explanation should be moved up).

**AC:** We meant wavelengths below 300 nm.

**MC:** We clarified this in line 103.

**RC3:** L106: Where wavelengths shifts just analyzed or were they actually corrected?

**AC:** The wavelength shifts were just analyzed. A dispersion test with HG and CD spectral lamps was done based on wavelengths shifts analyses in 2018 to improve the spectral measurements. The maximal wavelengths shifts were in the Brewer spectrum range (290-363 nm) from +0.02 to -0.08 nm before the application of new constants in 2018. Another two dispersions were collected during calibration on site with traveling Brewer in 2012 and 2016. Spectral data were not corrected backward.

**MC:** We specified this in lines 89–92.

**RC3:** L117: According to Figure 4, bottom, cloud cover at Marambio is the norm. So what TOC data were used on cloudy days?

**AC:** Only direct sun TOC observations were taken in account. Weather at Marambio, including cloud cover, is changeable, therefore we were able to get the direct sun (DS) readings on most days. Moreover, it is possible to take a DS measurements even through a thin cloud cover. The statistics shown in Fig. 4 have been calculated only from the studied cases, i.e. solar UV spectra paired with DS TOC measurements, ERA5 cloud cover, and albedo climatology.

**MC:** We clarified this in line 125.

**RC3:** L127: Is "cloudiness" equal to cloud cover? How is cloud cover expressed in this reanalysis data set? Is it in oktas or percent? Is it the fraction of clouds within the 0.25 x 0.25 degree pixel? Since Marambio is on a small island, are the ERA5 data representative? The optical thickness of clouds is at least as important as cloud cover. Hence, it seems that cloud cover from ERA5 reanalysis data is not a good quantity to assess the effects of clouds on UV radiation.

**AC:** In the study, the terms "cloudiness" and "cloud cover" were used as synonyms (the portion of sky covered by clouds), but to avoid confusion, we will be preferring "cloud cover" in the edited version of the manuscript.

The ERA5 dataset gives cloud cover in percentages (we will specify this in the manuscript). As for the spatial representativeness and the absence of cloud optical thickness, we agree that the dataset has its limitations, even though the cloud cover in the western part of Weddell Sea is quite consistent, affected

mostly by large-scale synoptic systems and processes, which are quite well represented in ERA5. Generally, it was the best cloud cover estimate available to us that covered the entire study period (for more information, please see the next point).

**MC:** The term "cloudiness" was changed to "cloud cover" throughout the manuscript. We also extended the information on the ERA5 dataset (lines 138 and 141–142).

**RC3:** L128: Regarding "the best correlation (r = 0.26) with". Best relative to what? Is it indeed r or should it be r^2? If it were r, r^2 would be 0.068, which means that only 6.8% of the variability in the CMF can be explained with ERA5 cloud cover data. This is a rather small fraction, confirming that the ERA5 reanalysis data are not very useful to describe cloud cover at Marambio, which also impacts the significance of results presented later. This should be discussed.

**AC:** Thank you for the notice. Unfortunately this is the case, most likely due to the omittance of cloud structure information in ERA5 data (i.e. both a cirrostratus and a nimbostratus would get the total cloud cover of 100 %, even when CMF would be greatly different). We only omitted the negative correlation sign, which we will of course correct in the manuscript. The other tested datasets were MERRA2 (r = -0.10) and OMI cloud fraction (r = -0.13). No direct observations from the site covering the entire studied period were available to us, so despite the limitations, related of course also to your previous comment regarding the data representativeness and cloud optical thickness, we decided to use the dataset in the study. This decision was supported by the fact that the inclusion of the ERA5 cloud cover dataset brought a statistically significant improvement in our ANN simulations (see Fig. A1a in the original manuscript's Appendix 1).

**MC:** The error in the correlation coefficient was corrected, other datasets (OMI, MERRA2) were mentioned, and the ERA5 dataset limitations were discussed (lines 138–142).

**RC3:** L130: "then the weighted mean using the clear-sky intensities as weights was derived for each spectrum." Is difficult to understand. Please rephrase and add detail.

**MC:** The sentence in lines 143–147 was rephrased as follows: "In order to calculate a single CMF value for the entire spectrum, a weighted mean was used, so the CMFs at each wavelength were multiplied by the corresponding modelled clear-sky UV irradiances, summed up and divided by the clear-sky irradiance integrated through all studied wavelengths."

**RC3:** L138: How representative is the OMUVB albedo climatology for the relative small footprint of the observation site considering that there is both ocean and land in the OMUVB pixel? Were there any albedo measurements at Marambio Base that could be used to validate the OMUVB albedo climatology?

**AC:** Unfortunately there were no albedo measurements at Marambio, which would cover the entire 2010–2020 study period, available to us, so we had to use the OMUVB climatology as the best possible albedo estimate. Of course, its representativeness is limited, as it covers a relatively large area and does not capture individual solid precipitation events and year-to-year changes, but we decided to include the dataset, as it presented a statistically significant improvement to our ANN model (see Fig. A1a in the manuscript's Appendix 1).

**RC3:** L153: You only considered the ANN and a regression model. Why didn't you use a physical correct radiative transfer model to detangle the effect of the different explanatory variables considering that all important input variables for such a model were available?

**AC:** Thank you for the note. It would indeed have been an interesting comparison and it would be great to see the differences between a radiative transfer and an ANN model. However, the problem was that not all the necessary input variables were available, especially when it comes to cloud cover.

**MC:** We briefly mentioned that due to the relatively simple datasets available to us it was not possible to perform radiative transfer modelling (line 173 – 174) and added a possible future research idea with the intercomparison of radiative transfer, ANN, and regression modelling (lines 448–449).

**RC3:** L158: What is the difference between testing and validation?

**AC:** A neural network uses a training subset for "training" (setting the quasi-optimal parameters of the neural network in iterative process). Despite the previous setting of the quasi-optimal complexity of the used network, however, the network may be overtrained in some cases. Therefore, a test subset is randomly selected from the input data before training and it is used to independently check possible overtraining of the network. When overtraining occurs, typically the network error on the training set decreases, but the error on the test set starts to increase. If the error on the test file grows for a certain number of training cycles (usually of the order of hundreds), the training is stopped ("early stopping") and the network with minimal error on the test subset is considered to be the final one. But, for the reasons mentioned above, the results are not independent of either the training or the test subsets, therefore a third (validation) subset is selected before training. This subset is not used for training of the network or checking for overtraining, but only to independently assess the performance of the final trained network. See also TIBCO Statistica® User's Guide, chapter „Network Generalization", cited above.

**MC:** The difference was briefly explained in lines 180–183.

**RC3:** L173: Delete: "while it was greater at shorter wavelengths, likely due to the smaller range of the data." This does not make sense to me. My guess is that $R^2$ is greater at small wavelengths because those depend more on ozone, which (anti)correlates much better with UV radiation than cloud cover.

**AC:** Thank you for the note. You may be right with the ozone dependency, as a clear, ozone-related Huggins band structure is visible in Fig. 1d.

**MC:** The interpretation attempt was deleted.

**RC3:** L183: What 9 months?

**AC:** We meant the months in which the B199 measurements were available (i.e. August to April).

**MC:** We clarified this in lines 216–217.

**RC3:** L197: Delete: "the UVB region". 330 nm is not in the UV-B range.

**MC:** We rephrased it to "short wavelengths" (line 232).

**RC3:** L211: I don't understand "relative differences gradient"

**MC:** The sentence containing this statement (lines 245–247) was rephrased as follows: "The smaller relative differences at short wavelengths (approximately 300–305 nm) in September and larger ones in October may be possibly attributed to the effect of variable ozone amount, as the ozone deficiency causes a relative increase in UV irradiance at short wavelengths."

**RC3:** L212: Regarding: "In October, when the UV irradiance is lower than the overall median, the difference decreases with decreasing wavelengths, while in November, when the median irradiance already exceeds the overall median, the difference increases more steeply." Looking at Figure 4c, the UV irradiance is higher (not lower) than then overall median in October and the difference increases (not decreases) with decreasing wavelength.

**AC:** Thank you for noticing this. We meant September and October here.

**MC:** We took out this sentence entirely as the rephrased sentence mentioned in the previous comment says the same thing.

**RC3:** L218: Karhu et al. (2003) or Koo et al. (2018) are not good references to cite here. I suggest to cite either the 2022 WMO ozone assessment (https://csl.noaa.gov/assessments/ozone/2022/) instead, which will be published soon, or the 1998 version.

**MC:** We replaced the citations by the new WMO ozone assessment (lines 251 and 694–695).

RC3: L225–229: While this is more or less correct, the explanation misses the point why both ozone minima and maxima can be observed in November. The main reason is that Marambio is sufficiently far away from the South Pole so that the station can be either below the ozone hole or outside its perimeter where ozone is typically high in November.

**MC:** This interpretation was added to the manuscript (lines 262–265).

**RC3:** L269: Attenuation of UV-B irradiance by thin (e.g., cirrus) clouds can be much less than 30%, in particular if snow is on the ground.

**MC:** We mentioned this in lines 305–306.

**RC3:** L296–300. I think you mean Fig. 7–9, not Fig. 7-10, in the paragraph.

**AC:** Yes, thank you for noticing this, it is a leftover from the previous iteration where we took out one figure.

**MC:** We corrected it accordingly (lines 333–335).

**RC3:** L299: The sentence "of the relationships of the explanatory variables and UV irradiance, given that all other variables are fixed to their monthly medians." is confusing. The sentence implies that there are other variables, in addition to explanatory variables and UV irradiance. Better: "Fig 7 shows the relationship between SZA and UV irradiance with the other explanatory variable fixed to their monthly mean. Similar relationships between TOC and UV irradiance and between cloud cover and UV irradiance are shown in Figs. 8 and 9, respectively."

**MC:** The suggestion was included in the manuscript (lines 335–337).

**RC3:** The captions of Figure 7–9 should be rearranged. I suggest (for Figure 7): "Modelled relationships between UV irradiance and SZA for different months. Panel (a) and (b) show the modelled UV irradiance at 305 nm (a) and 340 nm (b) as a function of SZA. Panels (c) and (d) show the change in spectral UV irradiance resulting from an increase in SZA by 1° in absolute (c) and relative (d) terms. Similar captions should be chosen for Figures 8 and 9.

**AC:** Thank you for the suggestion. As we are taking out panel (c) (see above), we rephrased the captions as follows (for Fig.7):

Modelled relationships between UV irradiance and SZA for different months. Panel (a) and (b) show the modelled UV irradiance at 305 nm (a) and 340 nm (b) as a function of SZA. Panel (c) shows the relative change in spectral UV irradiance resulting from an increase in SZA by 1°, calculated with reference to the monthly median spectra.

We kept the last part of the caption ("calculated with reference to the monthly median spectra"), as it was a suggestion of another referee to avoid possible confusion.

**MC:** Relevant changes were made in lines 741–753.

**RC3:** L328: The vertical profile becomes only important for SZA larger than about 75 degrees, when the Umkehr effect starts to become apparent. So the ozone profile does not plan a "substantial" role, unless the SZA is very large.

**MC:** The large SZA condition was included (line 368).

**RC3:** L340: Another important reasons why clouds have less influence in Antarctica compared to mid-latitude sites is that high surface albedo prevailing over the Antarctic continent greatly reduces cloud effects due to multiple scattering between clouds and the snow-covered surface. However, this may be of less importance at Marambio compared to the Antarctic continent.

**AC:** Thank you for the note. Albedo may play some role even at Marambio, as most precipitation falls in solid form and the sea is covered in sea ice in spring and fall.

**MC:** We included it in the interpretation (lines 388–391), together with the suggested reference of Nichol et al. (2003).

**RC3:** Figure 10: I find this figure rather confusing and not very helpful. It is obvious that measurement and model don't agree when the model is run with input parameters differing from the actual situation. I think the main point that the authors like to make is that low ozone values can be compensated by large SZAs (e.g., the red lines in Panel b). I would remove the figure from manuscript but let the authors decide.

**AC:** The figure is indeed not meant as a model validation. As we still find the results quite interesting and case-specific, we would like to present the figure differently, to make it more obvious it was not a model validation but rather a set of case studies showing how the studied variables can compensate the effects of one another (see below). We will also change the description in the results section, focusing more on the potential influence of selected variables and the interpretation of the real situation, trying to avoid any "model vs. observation" comments that may lead to confusion.

[Figure]

Fig. 4. A new proposed version of Fig. 10 for the manuscript. The grey area, rather than lines, should show the potential effect of the selected variable, while the red and blue lines show the real situation, i.e. the effect of all variables combined.

**MC:** We changed the interpretation (lines 397–416) and the figure itself (lines 755–759).

**RC3:** L405: I find it rather strange that the results of nine of the ten ANN models are basically identical but one (ANN01) is rather different. I would have expected a continuous distributions. It would be good if the authors could better explain what's different about ANN01. It seems to me that the training dataset of model ANN01 included outliers.

**AC:** The different models do not necessarily yield a continuous distribution. The random selection of training/testing/validation subsets and the random initialization of the networks before their training may introduce the influence of random factors into the process, making the model results rather discontinuous.

[Figure]

Fig. 5. Statistical characteristics of the spectral UV irradiance in training, testing and validation subsets of the individual ANN models.

As for the presence and distribution of potential outliers, Fig. 5 shows a very similar divisions between the three subsets in all ten ANN models. The differences between the subsets are visible just in the minimal and maximal values, but ANN01 does not show any specifications vastly different to the other nine models. Looking at the structure of the individual subsets' maximal values, it is very similar to for example ANN05, and therefore it does not indicate the presence of outliers. The difference of ANN01 must have been therefore caused by the random initialization of the network.

We assume that if we built another dozens of models (which is unfortunately not within our time scope and computing capacity), some of them, albeit probably a minority, would perform similarly to ANN01.

**MC:** We included Fig. 5 from this response in Appendix A (lines 760–762) to better illustrate the divisions of the datasets, and a short description was added (lines 457–458).

**RC3: \*\*\*Comments to language**

AC: Many thanks for all the language remarks. Where no additional comments are present, we accepted the suggestion and made respective changes in the manuscript. In the case of specific comments, we are noting the lines in which the changes are present in the revised version of the manuscript (for deletions and general comments, please see the file with tracked changes).

**\*\*Comments to language, general**

- The word "while" is used improperly throughout the document. "While" contrasts two different things, like "whereas". For example, one could say: irradiance at 305 nm is small while it is large at 340 nm. "While" is often used in the text instead of "for example", which is an incorrect meaning.
- Change "in average" to "on average" throughout.
- Don't use the expressions "high SZA" and "low SZA". These expressions are confusing because a reader associates "high" with the Sun being high in the sky while the opposite is the case. Use "large SZA" and "small SZA" instead.
- Change "Out of" to just "Of"
- Always place a space between value and unit. (It is done most of the time, but not always.)
- Change "Huggins belt" to "Huggins band"

**\*\*Comments to language, specific**

- L7: "assess response of spectral UV radiation to different atmospheric" > "to assess the response …" or better: "assess the dependence of spectral UV radiation on different…"
    - MC: line 7
- L8: "in southern polar" > "in the southern polar"; delete "individual unique" as this is obvious
    - MC: line 8
- L11: "the resolution of 0.5 nm." > a resolution of 0.5 nm." Also please double-check that this is correct for your instrument. To my knowledge, the spectral resolution of a Brewer MK III is 0.6 nm.
    - MC: line 11 (also "resolution" was changed to "sampling interval")
- L12: Define "TOC"
    - MC: line 12
- L13: decline > decrease
    - MC: line 13
- L13: Regarding the sentence: "Also TOC affects particularly the short wavelengths, while at 305 nm, a 10 DU decrease in TOC causes a 7–13 % increase in UV irradiance.", specify what you consider "short" wavelengths. The word "while" implies that 305 nm is a long wavelength because "while" is a word contrasting two different things. So you could say: TOC affects wavelengths below about 340 nm. For example, at 305 nm a 10 DU decrease in TOC causes a 7–13 % increase in UV irradiance."
    - MC: lines 14 – 15
- L17: Specify what "very high" means (e.g., the upper 10% of the distribution)
    - MC: lines 18–19
- L22: Delete "the thorough"

- L23: accounted > attributed
  - MC: line 25
- L25: "lead to melanoma" should refer to humans, not "other organisms"
  - AC: This part was removed based on the first "minor" comment.
- L26: "catalyses vitamin D creation" > "leads to the production of vit D"
  - AC: This part was removed based on the first "minor" comment.
- L29: Cite https://doi.org/10.1038/s41586-021-03737-3 in support of "terrestrial plant productivity"
  - MC: line 27
- L31: "short-wavelength UV irradiance" > "from reaching the surface"
  - MC: line 29
- L32: "in the 1980s" > "in 1985"
  - MC: line 30
- L33: "many events have taken place to eliminate the ozone depletion," > " many efforts have been made to reduce ozone depletion"
  - MC: line 31
- L33: "the 1987 Montreal Protocol and its numerous amendments" > "through the passing of the Montreal Protocol in 1987 and subsequent amendments to this landmark treaty."
  - MC: lines 31–32
- L35: "in September" > "for the month of September"
  - AC: This part was removed based on the first "minor" comment.
- L37: "ozone hole still" > "the ozone hole still"
  - MC: line 35
- L38: Delete "the rare"
- L43: Remove "as it is illustrated even by the possible division at 315 nm" The division is indeed somewhat arbitrary, but the publications cited do not "illustrate" this.
  - MC: The sentence (lines 39–40) was rephrased as follows: "However, this division is semi-arbitrary, as there is no clear, physically defined transition between UVA and UVB bands (e.g., Diffey, 1990; Juzeniene et al., 2011)."
- L48: Indicate that NH means nitrogen hydrogen
  - MC: line 44
- L49: "but due to their respective abundance" > "but due to the difference in their respective abundances"
  - MC: lines 45–46
- L53: precise > large
  - MC: line 49
- L54: "the solar UV spectra" > "solar UV spectra"
  - MC: line 50
- L55: delete "already"
- L57: Delete: "especially without further processing"
- L58: Delete "the UV Index" (The UV Index is not an action spectrum)
- L66: "artificial neural network modelling (ANN)" > "artificial neural network (ANN) modelling"
  - MC: line 64

- L69: Avoid strings of nouns: "solar UV spectral irradiance observation" > "observations of spectral solar UV irradiance"
  - AC: This part was removed based on the first "minor" comment.
- L79: doesn't > does not
  - MC: line 76
- L80: "all year long" > "at any time of the year"
  - MC: line 77
- L88: "has been performed" > "was performed"
  - MC: lines 85–86
- L89: travel > travelling
  - MC: line 89
- L92: "of HG and CD > "of Hg and Cd" or better "mercury and cadmium"
  - MC: line 90
- L102: "weak transmitted information affected" > "spectral UV irradiance at the Earth surface, which is affected"
  - MC: line 105
- L107: "The 23 260 of the spectra," > "The subset of 23260 spectra that passed quality control were used for this study and paired with explanatory variables" (if that's what you mean)
  - AC: Not precisely, as far more spectra (over 40 000) passed the quality control but only the 23 260 of them were successfully paired with explanatory variables based on the criteria explained further (e.g., TOC measurement taken no less than 60 minutes before or after the spectral observation etc.).
  - MC: To avoid confusion, we rephrased the sentence in lines 110–112 as follows: "The subset of 23 260 spectra that passed quality control and was successfully paired with explanatory variables based on selected criteria (see Section 2.3), was used for this study."
- L109: "It can be seen there were several data gaps, of which the longest occurred " > "It can be seen that there were several data gaps, the longest of which occurred"
  - MC: lines 112–113
- L113: "has been paired with following explanatory" > "was paired with the following explanatory"
  - MC: line 120
- L115: "belong to" > "are"
  - MC: line 122
- L119: "Therefore, more solar spectra could be matched" > "Therefore, several solar spectra were sometimes matched"
  - MC: line 127
- L120: "provided it was taken within the 60 minute time distance and it was the closest observation." > "provided that they were taken within 60 minutes of the closest ozone observation"
  - MC: To make it more clear, we rephrased this as follows: "provided the ozone observation was taken within 60 minutes of the spectral measurement and there was no other ozone observation with a shorter interval from the spectral measurement" (lines 128–129).

- L134: "(further)." > "(see Section XX)."
  - AC: This is in the same section, actually the very next paragraph.
  - MC: We rephrased it to "see further below" (line 149).
- L153: "further in this section)." > "further below"
  - MC: line 170
- L162: "more information is included in Appendix" > "see Appendix"
  - MC: line 194
- L166: "However, even in" > "However, even for"; "carried on: median bias values" > carried out: the median bias between measurement and model values"
  - MC: line 198
- L167: delete "then they"
- L168: "tackled" > "removed"
  - MC: line 200
- L169: "fitted between" > "agreed to within"
  - MC: line 201
- L170: "from approximately 310 nm," > "for wavelengths longer than approximately 310 nm,"; "modelled data was within approximately" > "measured and modelled data agreed within approximately"
  - MC: lines 202–203
- L172: "determination coefficient" > "coefficient of determination"
  - MC: line 204
- L174: "shared variance" is an uncommon term. Use "coefficient of determination" instead.
  - MC: we used "R-squared" (line 206), which we use for coefficient of determination (a suggestion from another anonymous referee).
- L177: "Out of the four explanatory variables, always only a single one was left to its original value, while the three other variables were fixed to their monthly medians" > "Of the four explanatory variables, one was selected and retained at its original value while the three other variables were fixed to their monthly medians. The procedure was then repeated for each of the four variables."
  - MC: The albedo effects study was taken out (see previous iteration), so we rephrased this as follows: "The procedure repeated for each of the variables except albedo, whose effects were not studied further as its climatology was used" (lines 210–211).
- L195: "increased" > "increases"
  - MC: line 228
- L200 "of the Sun and the studied site" > "of the Sun at Marambio Base"
  - MC: line 234
- L202: "increase in median irradiance varies" > "increase in median irradiance per nm varies"
  - MC: line 237
- L205: Delete "precise"
- L214: "The wave-like Huggins belt" > "The wave-like structure in the Huggins band"
  - MC: lines 247–248
- L217: "for Antarctic" > "for the Antarctic"
  - MC: line 250
- L222: "conditioned" > "depend both on"
  - MC: line 255

- L270: Delete "likely"
- L277: "which may lead to its decline compared to snow cover or glaciated areas" > "which will lead to lower UV radiation compared to snow cover or glaciated areas"
  - MC: line 314
- L307: "indirectly proportional" > "inversely proportional"
  - MC: line 344
- L385: "wavelengths, when at" > "wavelengths. At"
  - MC: line 432
- L390: "at very short wavelengths is most visible." > "is most visible at very short wavelengths."
  - MC: lines 437–438

---

## Author Response (AR3)

**Author response to referee comments**

1. Referee comments (RC3) are in **BLACK**
2. Authors' answers (AC) are in **BLUE**
3. Changes in the manuscript (MC) are in **PURPLE**; the lines refer not to the original, but to the revised version of the manuscript.

**Author's response to comments of Referees #3 and #4**

**AC:** We thank the anonymous referees for the review of the manuscript. The minor comment by Referee #3 is addressed below, alongside the changes that were made to the manuscript based on this comment.

**RC3:** The titles of the y-axis on the paper's figures still read "Irradiance" instead of "Spectral irradiance" even though the units are now (correctly) those of spectral irradiance.

**AC:** Thank you for the note, we changed all the required axis titles accordingly.

**MC:**

- Fig. 4 (line 730): y-axis title was changed in panels (a) and (b)
- Fig. 5 (line 733): legend of panel (a) was changed
- Fig. 7 (line 741): y-axis title was changed in panel (c)
- Fig. 8 (line 746): y-axis title was changed in panel (c)
- Fig. 9 (line 750): y-axis title was changed in panel (c)
- Fig. A1 (line 760): y-axis title was changed in all ten panels